# Election Coding for Distributed Learning: Protecting SignSGD against Byzantine Attacks

**Jy-yong Sohn**
jysohn1108@kaist.ac.kr

**Dong-Jun Han**
djhan93@kaist.ac.kr

**Beongjun Choi**
bbzang10@kaist.ac.kr

**Jaekyun Moon**
jmoon@kaist.edu

School of Electrical Engineering,
Korea Advanced Institute of Science and Technology (KAIST)

## Abstract

Current distributed learning systems suffer from serious performance degradation under Byzantine attacks. This paper proposes ELECTION CODING, a coding-theoretic framework to guarantee Byzantine-robustness for distributed learning algorithms based on signed stochastic gradient descent (SignSGD) that minimizes the worker-master communication load. The suggested framework explores new information-theoretic limits of finding the majority opinion when some workers could be attacked by adversary, and paves the road to implement robust and communication-efficient distributed learning algorithms. Under this framework, we construct two types of codes, random Bernoulli codes and deterministic algebraic codes, that tolerate Byzantine attacks with a controlled amount of computational redundancy and guarantee convergence in general non-convex scenarios. For the Bernoulli codes, we provide an upper bound on the error probability in estimating the signs of the true gradients, which gives useful insights into code design for Byzantine tolerance. The proposed deterministic codes are proven to perfectly tolerate arbitrary Byzantine attacks. Experiments on real datasets confirm that the suggested codes provide substantial improvement in Byzantine tolerance of distributed learning systems employing SignSGD.

## 1  Introduction

The modern machine learning paradigm is moving toward parallelization and decentralization [4, 12, 18] to speed up the training and provide reliable solutions to time-sensitive real-world problems. There has been extensive work on developing distributed learning algorithms [1, 16, 21, 22, 24, 25] to exploit large-scale computing units. These distributed algorithms are usually implemented in parameter-server (PS) framework [17], where a central PS (or master) aggregates the computational results (e.g., gradient vectors minimizing empirical losses) of distributed workers to update the shared model parameters. In recent years, two issues have emerged as major drawbacks that limit the performance of distributed learning: *Byzantine attacks* and *communication burden*.

Nodes affected by Byzantine attacks send arbitrary messages to PS, which would mislead the model updating process and severely degrade learning capability. To counter the threat of Byzantine attacks, much attention has been focused on robust solutions [2, 13, 14]. Motivated by the fact that a naive linear aggregation at PS cannot even tolerate Byzantine attack on a single node, the authors of [7, 10, 31] considered median-based aggregation methods. However, as data volume and the number of workers increase, computing the median involves a large cost [7] far greater than the cost

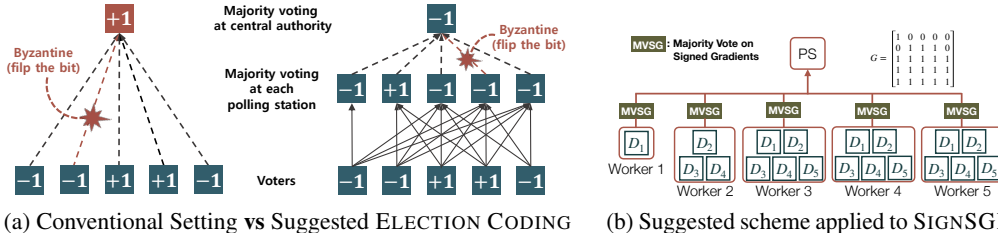

(a) Conventional Setting **vs** Suggested ELECTION CODING     (b) Suggested scheme applied to SIGNSGD

Figure 1: Use of coding to protect the majority vote. (a) The conventional scheme (without coding) is vulnerable to Byzantine attacks, while the suggested scheme is not. In the suggested scheme, each polling station gathers the votes of a subset of voters, and sends the majority value to the master. (b) In the real setup, voters are like data partitions $D_i$ while polling stations represent workers.

for batch gradient computations. Thus, recent works [9, 23] instead suggested redundant gradient computation that tolerates Byzantine attacks.

Another issue is the high communication burden caused by transmitting gradient vectors between PS and workers for updating network models. Regarding this issue, the authors of [3, 5, 6, 15, 19, 27–29] considered quantization of real-valued gradient vectors. The signed stochastic gradient descent method (SIGNSGD) suggested in [5] compresses a real-valued gradient vector $\boldsymbol{g}$ into a binary vector $\mathrm{sign}(\boldsymbol{g})$, and updates the model using the 1-bit compressed gradients. This scheme minimizes the communication load from PS to each worker for transmitting the aggregated gradient. A further variation called SIGNSGD WITH MAJORITY VOTE (SIGNSGD-MV) [5, 6] also applies 1-bit quantization on gradients communicated from each worker to PS in achieving minimum master-worker communication in both directions. These schemes have been shown to minimize the communication load while maintaining the SGD-level convergence speed in general non-convex problems. A major issue that remains is the lack of Byzantine-robust solutions suitable for such communication-efficient learning algorithms.

In this paper, we propose ELECTION CODING, a coding-theoretic framework to make SIGNSGD-MV [5] highly robust to Byzantine attacks. In particular, we focus on estimating the next step for model update used in [5], i.e., the *majority* voting on the signed gradients extracted from $n$ data partitions, under the scenario where $b$ of the $n$ worker nodes are under Byzantine attacks.

Let us illustrate the concept of the suggested framework as a voting scenario where $n$ people vote for either one candidate $(+1)$ or the other $(-1)$. Suppose that each individual must send her vote directly to the central election commission, or a master. Assume that frauds can happen during the transmittal of votes, possibly flipping the result of a closely contested election, as a single fraud did in the first example of Fig. 1a. A simple strategy can effectively combat this type of voting fraud. First set up multiple polling stations and let each person go to multiple stations to cast her ballots. Each station finds the majority vote of its poll and sends it to the central commission. Again a fraud can happen as the transmissions begin. However, as seen in the second example of Fig. 1a, the single fraud was not able to change the election result, thanks to the built-in redundancy in the voting process. In this example, coding amounts to telling each voter to go to which polling stations. In the context of SignSGD, the individual votes are like the locally computed gradient signs that must be sent to the PS, and through some ideal redundant allocation of data partitions we wish to protect the integrity of the gathered gradient computation results under Byzantine attacks on locally computed gradients.

**Main contributions:** Under this suggested hierarchical voting framework, we construct two ELECTION coding schemes: random Bernoulli codes and deterministic algebraic codes. Regarding the random Bernoulli codes, which are based on arbitrarily assigning data partitions to each node with probability $p$, we obtain an upper bound on the error probability in estimating the sign of the true gradient. Given $p = \Theta(\sqrt{\log(n)/n})$, the estimation error vanishes to zero for arbitrary $b$, under the asymptotic regime of large $n$. Moreover, the convergence of Bernoulli coded systems are proven in general non-convex optimization scenarios. As for the deterministic codes, we first obtain the necessary and sufficient condition on the data allocation rule, in order to accurately estimate the majority vote under Byzantine attacks. Afterwards, we suggest an explicit coding scheme which achieves perfect Byzantine tolerance for arbitrary $n, b$.

Finally, the mathematical results are confirmed by simulations on well-known machine learning architectures. We implement the suggested coded distributed learning algorithms in PyTorch, and deploy them on Amazon EC2 using Python with MPI4py package. We trained RESNET-18 using CIFAR-10 dataset as well as a logistic regression model using Amazon Employee Access dataset.

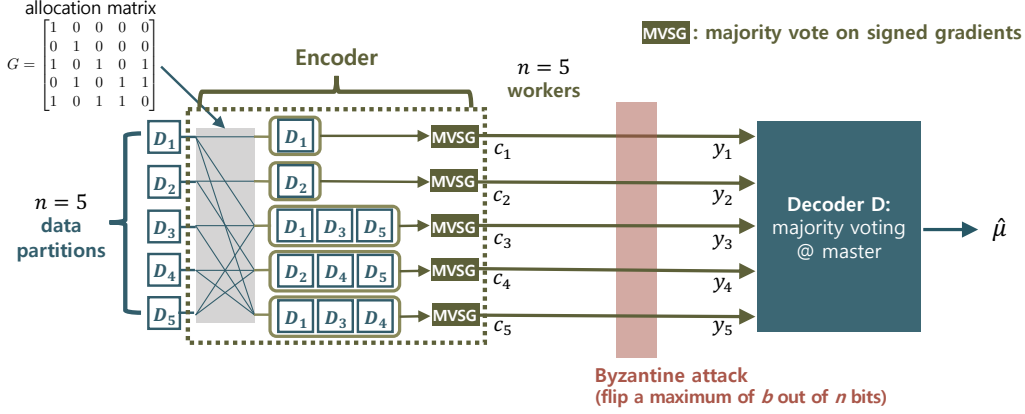

Figure 2: A formal description of the suggested ELECTION CODING framework for estimating the majority opinion $\mu$. This framework is applied for each coordinate of the model parameter $\boldsymbol{w} \in \mathbb{R}^d$ in a parallel manner.

The experimental results confirm that the suggested coded algorithm has significant advantages in tolerating Byzantines compared to the conventional uncoded method, under various attack scenarios.

**Related works:** The authors of [9] suggested a coding-theoretic framework DRACO for Byzantine-robustness of distributed learning algorithms. Compared to the codes in [9], our codes have the following two advantages. First, our codes are more suitable for SIGNSGD setup (or in general compressed gradient schemes) with limited communication burden. The codes proposed in [9] were designed for real-valued gradients, while our codes are intended for quantized/compressed gradients. Second, the random Bernoulli codes suggested in this paper can be designed in a more flexible manner. The computational redundancy $r = 2b + 1$ of the codes in [9] linearly increases, which is burdensome for large $b$. As for Bernoulli codes proposed in this paper, we can control the redundancy by choosing an appropriate connection probability $p$. Simulation results show that our codes having a small expected redundancy of $\mathbb{E}[r] = 2, 3$ enjoy significant gain compared to the uncoded scheme for various $n, b$ settings. A recent work [23] suggested a framework DETOX which combines two existing schemes: computing redundant gradients and robust aggregation methods. However, DETOX still suffers from a high computational overhead, since it relies on the computation-intensive geometric median aggregator. For communicating 1-bit compressed gradients, a recent work [6] analyzed the Byzantine-tolerance of the naive SIGNSGD-MV scheme. This scheme can only achieve a limited accuracy as $b$ increases, whereas the proposed coding schemes can achieve high accuracy in a wide range of $b$ as shown in the experimental results provided in Section 5. Moreover, as proven in Section 4, the suggested deterministic codes achieve the ideal accuracy of $b = 0$ scenario, regardless of the actual number of Byzantines in the system.

**Notations:** The sum of elements of vector $\boldsymbol{v}$ is denoted as $\|\boldsymbol{v}\|_0$. Similarly, $\|\mathbf{M}\|_0$ represents the sum of elements of matrix $\mathbf{M}$. The sum of the absolute values of the vector elements is denoted as $\|\boldsymbol{v}\|_1$. An $n \times n$ identity matrix is denoted as $\mathbf{I}_n$. The set $\{1, 2, \ldots, n\}$ is denoted by $[n]$. An $n \times k$ all-ones matrix is denoted as $\mathbb{1}_{n \times k}$.

## 2 Suggested framework for Byzantine-robust distributed learning

### 2.1 Preliminary: SIGNSGD WITH MAJORITY VOTE (SIGNSGD-MV)

Here we review SIGNSGD WITH MAJORITY VOTE [5, 6] applied to distributed learning setup with $n$ workers, where the goal is to optimize model parameter $\boldsymbol{w} \subseteq \mathbb{R}^d$. Assume that each training data is sampled from distribution $\mathcal{D}$. We divide the training data into $n$ partitions, denoted as $\{D_j\}_{j \in [n]}$. Let $\boldsymbol{g} = [g^{(1)}, \cdots, g^{(d)}]$ be the gradient calculated when the whole training data is given. Then, the output of the stochastic gradient oracle for an input data point $x$ is denoted by $\tilde{\boldsymbol{g}}(x) = [\tilde{g}^{(1)}(x), \cdots, \tilde{g}^{(d)}(x)]$, which is an estimate of the ground-truth $\boldsymbol{g}$. Since processing is identical across $d$ coordinates or dimensions, we shall simply focus on one coordinate $k \in [d]$, with the understanding that encoding/decoding and computation are done in parallel across all coordinates. The superscript will be dropped unless needed for clarity. At each iteration, $B$ data points are selected for each data partition. We denote the set of data points selected for the $j^{\text{th}}$ partition as $\mathcal{B}_j \subseteq D_j$, satisfying $|\mathcal{B}_j| = B$. The output of the stochastic gradient oracle for $j^{\text{th}}$ partition is expressed as $\tilde{g}_j = \sum_{x \in \mathcal{B}_j} \tilde{g}(x)/B$. For a specific coordinate, the set of gradient elements computed for $n$ data partitions is denoted as $\tilde{\boldsymbol{g}} = [\tilde{g}_1, \cdots, \tilde{g}_n]$. The sign of the gradient is represented

as $\boldsymbol{m} = [m_1, \cdots, m_n]$ where $m_j = \mathrm{sign}(\tilde{g}_j) \in \{+1, -1\}$. We define the *majority opinion* as $\mu = \mathrm{maj}(\boldsymbol{m})$, where $\mathrm{maj}(\cdot)$ is the *majority* function which outputs the more frequent element in the input argument. At time $t$, SIGNSGD-MV updates the model as $w_{t+1} = w_t - \gamma\mu$, where $\gamma$ is the learning rate.

## 2.2 Proposed ELECTION CODING framework

The suggested framework for estimating the majority opinion $\mu$ is illustrated in Fig. 2. This paper suggests applying codes for allocating data partitions into worker nodes. The data allocation matrix $\mathbf{G} \in \{0, 1\}^{n \times n}$ is defined as follows: $G_{ij} = 1$ if data partition $j$ is allocated to node $i$, and $G_{ij} = 0$ otherwise. We define $P_i = \{j : G_{ij} = 1\}$, the set of data partitions assigned to node $i$. Given a matrix $\mathbf{G}$, the computational redundancy compared to the uncoded scheme is expressed as $r = \|\mathbf{G}\|_0/n$, the average number of data partitions handled by each node. Note that the uncoded scheme corresponds to $\mathbf{G} = \mathbf{I}_n$. Once node $i$ computes $\{m_j\}_{j \in P_i}$ from the assigned data partitions, it generates a bit $c_i = E_i(\boldsymbol{m}; \mathbf{G}) = \mathrm{maj}(\{m_j\}_{j \in P_i})$ using encoder $E_i$. In other words, node $i$ takes the majority of the signed gradients obtained from partitions $D_j$ observed by the node. We denote the $n$ bits generated from worker nodes by $\boldsymbol{c} = [c_1, \cdots c_n]$. After generating $c_i \in \{+1, -1\}$, node $i$ transmits

$$y_i = \begin{cases} \mathcal{X}, & \text{if node } i \text{ is a Byzantine[1]node} \\ c_i, & \text{otherwise} \end{cases} \tag{1}$$

to PS, where $\mathcal{X} \in \{c_i, -c_i\}$ holds since each node is allowed to transmit either $+1$ or $-1$. We denote the number of Byzantine nodes as $b = n\alpha$, where $\alpha$ is the portion corresponding to adversaries. PS observes $\boldsymbol{y} = [y_1, \cdots, y_n]$ and estimates $\mu$ using a decoding function $D$. Using the output $\hat{\mu}$ of the decoding function, PS updates the model as $\omega_{t+1} = \omega_t - \gamma\hat{\mu}$.

There are three design parameters which characterize the suggested framework: the data allocation matrix $\mathbf{G}$, the encoder function $E_i$ at worker $i \in [n]$, and the decoder function $D$ at the master. In this paper, we propose low-complexity hierarchical voting where both the encoder/decoder are majority voting functions. Under this suggested framework, we focus on devising $\mathbf{G}$ which tolerates Byzantine attacks on $b$ nodes. Although this paper focuses on applying codes for SignSGD, the proposed idea can easily be extended to multi-bit quantized SGD with a slight modification: instead of finding the majority value at the encoder/decoder, we take the average and then round it to the nearest quantization level.

# 3 Random Bernoulli codes

We first suggest random Bernoulli codes, where each node randomly selects each data partition with connection probability $p$ independently. Then, $\{G_{ij}\}$ are independent and identically distributed Bernoulli random variables with $G_{ij} \sim Bern(p)$. The idea of random coding is popular in coding and information theory (see, for example, [20]) and has also been applied to computing distributed gradients [8], but random coding for tolerating Byzantines in the context of SignSGD is new and requires unique analysis. Note that depending on the given Byzantine attack scenario, flexible code construction is possible by adjusting the connection probability $p$. Before introducing theoretical results on random Bernoulli codes, we clarify the assumptions used for the mathematical analysis.

We assume several properties of loss function $f$ and gradient $g = \nabla f$, summarized as below. Assumptions 1, 2, 3 are commonly used in proving the convergence of general non-convex optimization, and Assumption 4 is used and justified in previous works on SIGNSGD [5,6].

**Assumption 1.** *For arbitrary $x$, the loss value is bounded as $f(x) \geq f^\star$ for some constant $f^\star$.*

**Assumption 2.** *For arbitrary $x, y$, there exists a vector of Lipschitz constants $\boldsymbol{L} = [L^{(1)}, \cdots, L^{(d)}]$ satisfying $|f(y) - [f(x) + g(x)^T(y - x)]| \leq \frac{1}{2}\sum_i L^{(i)}(y^{(i)} - x^{(i)})^2$.*

**Assumption 3.** *The output of the stochastic gradient oracle is the unbiased estimate on the ground-truth gradient, where the variance of the estimate is bounded at each coordinate. In other words, for arbitrary data point $x$, we have $\mathbb{E}[\tilde{g}(x)] = g$ and $\mathbb{E}[(\tilde{g}(x) - g)^2] \leq \sigma^2$.*

**Assumption 4.** *The component of the gradient noise for data point $x$, denoted by $\tilde{g}(x) - g$, has a unimodal distribution that is symmetric about the zero mean, for all coordinates and all $x \sim \mathcal{D}$.*

### 3.1 Estimation error bound

In this section, we measure the probability of the PS correctly estimating the sign of the true gradient, under the scenario of applying suggested random Bernoulli codes. To be specific, we compare $\text{sign}(g)$, the sign of the true gradient, and $\hat{\mu}$, the output of the decoder at the master. Before stating the first main theorem which provides an upper bound on the estimation error $\mathbb{P}(\hat{\mu} \neq \text{sign}(g))$, we start from finding the statistics of the mini-batch gradient $\tilde{g}_j$ obtained from data partition $D_j$, the proof of which is in Supplementary Material.

**Proposition 1** (Distribution of batch gradient). *The mini-batch gradient $\tilde{g}_j$ for data partition $D_j$ follows a unimodal distribution that is symmetric around the mean $g$. The mean and variance of $\tilde{g}_j$ are given as $\mathbb{E}[\tilde{g}_j] = g$, and $var(\tilde{g}_j) = \mathbb{E}[(\tilde{g}_j - g)^2] \leq \bar{\sigma}^2 := \sigma^2/B$.*

**Definition 1.** *Define $S^{(k)} = \left| g^{(k)} \right| / \bar{\sigma}^{(k)}$ as the signal-to-noise ratio (SNR) of the stochastic gradient observed at each mini-batch, for coordinate $k \in [d]$.*

Now, we measure the estimation error of random Bernoulli codes. Recall that we consider a hierarchical voting system, where each node makes a local decision on the sign of gradient, and the PS makes a global decision by aggregating the local decisions of distributed nodes. We first find the local estimation error $q$ of a Byzantine-free (or intact) node $i$ as below, which is proven in Supplementary Material.

**Lemma 1** (Impact of local majority voting). *Suppose $p \geq p^\star = 2\sqrt{\frac{C \log(n)}{n}}$ for some $C > 0$. Then, the estimation error probability of a Byzantine-free node $i$ on the sign of gradient along a given coordinate is bounded:*

$$q = \mathbb{P}(c_i \neq \text{sign}(g)) \leq q^\star := 2 \cdot \max \left\{ \frac{2}{n^{2C}}, e^{-\sqrt{Cn \log(n)} \frac{S^2}{2(S^2+4)}} \right\}, \tag{2}$$

*where the superscripts on $g$ and $S$ that point to the coordinate are dropped to ease the nontation. As the connection probability $p$ increases, we can take a larger $C$, thus the local estimation error bound decreases. This makes sense because each node tends to make a correct decision as the number of observed data partitions increases.*

Now, consider the scenario with $b = n\alpha$ Byzantine nodes and $n(1 - \alpha)$ intact nodes. From (1), an intact node $i$ sends $y_i = c_i$ for a given coordinate. Suppose all Byzantine nodes send the reverse of the true gradient, i.e., $y_i = -\text{sign}(g)$, corresponding to the worst-case scenario which maximizes the global estimation error probability $P_{\text{global}} = \mathbb{P}(\hat{\mu} \neq \text{sign}(g))$. Under this setting, the global error probability at PS is bounded as below, the proof of which is given in Supplementary Material.

**Theorem 1** (Estimation error at master). *Consider the scenario with connection probability $p = \Theta(\sqrt{\log(n)/n})$ as in Lemma 1. Suppose the portion of Byzantine-free nodes satisfies*

$$1 - \alpha > \frac{(\sqrt{\log(\Delta)/n} + \sqrt{\log(\Delta)/n + 4u_{min}^\star})^2}{8(u_{min}^\star)^2} \tag{3}$$

*for some $\Delta > 2$, where $u_{min}^\star = 1 - \max_k\{q^{(k)\star}\}$ is a lower bound on the local estimation success probability. Then, for each coordinate, the global estimation error at PS is bounded as*

$$P_{global} = \mathbb{P}(\hat{\mu} \neq \text{sign}(g)) < 1/\Delta.$$

This theorem specifies the sufficient condition on the portion of Byzantine-free nodes that allows the global estimation error smaller than $1/\Delta$. Suppose $n$ and $p$ are given, thus $u_{min}^\star$ is fixed. As $\Delta$ increases, the right-hand-side of (3) also increases. This implies that to have a smaller error bound (i.e., larger $\Delta$), the portion of Byzantine-free nodes needs to be increased, which makes sense.

**Remark 1.** *In the asymptotic regime of large $n$, the condition (3) reduces to $1 - \alpha > \frac{1}{2u_{min}^\star} \rightarrow \frac{1}{2}$ for arbitrary $\Delta$, since $q^\star \rightarrow 0$ as shown in (2). This implies that the global estimation error vanishes to zero even in the extreme case of having maximum Byzantine nodes $b = n/2$, provided that the number of nodes $n$ is in the asymptotic regime and the connection probability satisfies $p = \Theta(\sqrt{\log(n)/n})$.*

Remark 1 states the behavior of global error bound for asymptotically large $n$. In a more practical setup with, say, a Byzantines portion of $\alpha = 0.2$, the batch size of $B = 128$, the connection factor of $C = 1$, and the SNR of $|g|/\sigma = 1$, the global error is bounded as $P_{\text{global}} < 0.01$ given $n \geq 40$.

---

**Algorithm 1** Data allocation matrix $\mathbf{G}$ satisfying perfect $b-$Byzantine tolerance ($0 < b < \lfloor n/2 \rfloor$)

---

**Input:** Number of nodes $n$, number of Byzantine nodes $b$.
**Output:** Data allocation matrix $\mathbf{G} \in \{0,1\}^{n \times n}$ that achieves perfect $b-$Byzantine tolerance.
**Initialize:** Define $s = \frac{n-1}{2} - b$ and $L = \lfloor \frac{n-(2b+1)}{2(b+1)} \rfloor + 1$. Initialize $\mathbf{G}$ as the all-zero matrix.
**Step 1:** Set the top left $s$-by-$s$ submatrix of $\mathbf{G}$ as identity matrix, i.e., $\mathbf{G}(1:s, 1:s) = \mathbf{I}_s$.
**Step 2:** Set the bottom $(n - s - L)$ rows as the all-one matrix, i.e., $\mathbf{G}(s + L + 1 : n, :) = \mathbb{1}_{(n-s-L) \times n}$.
**Step 3:** Fill in the matrix $\mathbf{A} = \mathbf{G}(s+1:s+L, s+1:n)$ as follows: Insert $2b + 1$ ones on each row by shifting the location by $b + 1$, i.e., $\mathbf{A}(l, (l-1)(b+1) + (1 : 2b+1)) = \mathbb{1}_{1 \times (2b+1)}$ for $l = 1, \cdots, L$.

---

### 3.2 Convergence analysis

The convergence of the suggested scheme can be formally stated as follows:

**Theorem 2** (Convergence of the Bernoulli-coded SIGNSGD-MV). *Suppose that Assumptions 1, 2, 3, 4 hold and the portion of Byzantine-free nodes satisfies* (3) *for $\Delta > 2$. Apply the random Bernoulli codes with connection probability $p = \Theta(\sqrt{\log(n)/n})$ on SignSGD-MV, and run SignSGD-MV for $T$ steps with an initial model $\boldsymbol{w}_0$. Define the learning rate as $\gamma(T) = \sqrt{\frac{f(\boldsymbol{w}_0) - f^*}{\|\boldsymbol{L}\|_1 T}}$. Then, the suggested scheme converges as $T$ increases in the sense*

$$\frac{1}{T} \sum_{t=0}^{T-1} \mathbb{E}\left[\|g(\boldsymbol{w}_t)\|_1\right] \leq \frac{3\|\boldsymbol{L}\|_1}{2(1 - 2/\Delta)}\gamma(T) \to 0 \quad as \ T \to \infty.$$

A full proof is given in Supplementary Material. Theorem 2 shows that the average of the absolute value of the gradient $g(\boldsymbol{w}_t)$ converges to zero, meaning that the gradient itself becomes zero eventually. This implies that the algorithm converges to a stationary point.

### 3.3 Few remarks regarding random Bernoulli codes

**Remark 2.** *We have assumed that the number of data partitions $k$ is equal to the number of workers $n$. What if $n \neq k$? In this case, the required connection probability for Byzantine tolerance in Lemma 1 behaves as $p \sim \sqrt{\log(k)/k}$. Since the computational load at each worker is proportional to $p$, the load can be lowered by increasing $k$. However, the local majority vote becomes less accurate with increasing $k$ since each worker uses fewer data points, i.e., parameter $S$ in Definition 1 deteriorates.*

**Remark 3.** *What is the minimal code redundancy while providing (asymptotically) perfect Byzantine tolerance? From the proof of Lemma 1, we see that if the connection probability (which is proportional to redundancy) can be expressed as $p = f(k)/\sqrt{k}$ with $f(k)$ being a growing function of $k$, the number of data partition, then the random codes achieve full Byzantine protection (in the asymptotes of $k$). Our choice $p \sim \sqrt{\log(k)/k}$ meets this, although it is not proven whether this represents minimal redundancy.*

**Remark 4.** *The worker loads of random Bernoulli codes are non-uniform, but the load distribution tends to uniform as $n$ grows, due to concentration property of binomial distribution.*

## 4 Deterministic codes for perfect Byzantine tolerance

We now construct codes that guarantee perfect Byzantine tolerance. We say the system is *perfect* $b$-Byzantine tolerant if $\mathbb{P}(\hat{\mu} \neq \mu) = 0$ holds on all coordinates[2] under the attack of $b$ Byzantines. We devise codes that enable such systems. Trivially, no probabilistic codes achieve this condition, and thus we focus on *deterministic* codes where the entries of the allocation matrix $\mathbf{G}$ are fixed. We use the notation $\mathbf{G}(i, :)$ to represent $i^{th}$ row of $\mathbf{G}$. We assume that the number of data partitions $\|\mathbf{G}(i, :)\|_0$ assigned to an arbitrary node $i$ is an odd number, to avoid ambiguity of the majority function at the encoder. We further map the sign values $\{+1, -1\}$ to binary values $\{1, 0\}$ for ease of presentation. Define

$$S_v(\boldsymbol{m}) = \{i \in [n] : \boldsymbol{m}^T \mathbf{G}(i, :) \geq v, \|\mathbf{G}(i, :)\|_0 = 2v - 1\}, \tag{4}$$

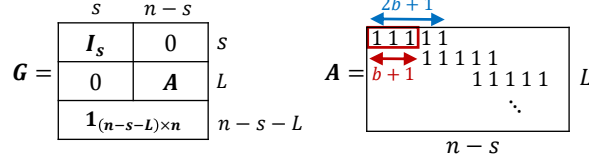

Figure 3: The structure of generator matrix $\mathbf{G}$ devised in Algorithm 1

which is the set of nodes having at least $v$ partitions with $m_j = 1$, out of $2v - 1$ allocated data partitions. Note that we have $c_i = \mathrm{maj}(\{m_j\}_{j \in P_i}) = 1$ iff $i \in S_v(\boldsymbol{m})$ holds for some $v$.

Before providing the explicit code construction rule (i.e., data allocation matrix $\mathbf{G}$), we state the necessary and sufficient condition on $\mathbf{G}$ to achieve *perfect $b$-Byzantine tolerance*.

**Lemma 2.** *Consider using a data allocation matrix* $\mathbf{G}$. *The system is perfect $b-$Byzantine tolerant if and only if* $\sum_{v=1}^{\lfloor n/2 \rfloor} |S_v(\boldsymbol{m})| \leq \lfloor \frac{n}{2} \rfloor - b$ *for all vectors* $\boldsymbol{m} \in \{0,1\}^n$ *having weight* $\|\boldsymbol{m}\|_0 = \lfloor n/2 \rfloor$.

*Proof.* The formal proof is in Supplementary Material, and here we just give an intuitive sketch. Recall that the majority opinion is $\mu = 0$ when $\boldsymbol{m}$ has weight $\|\boldsymbol{m}\|_0 = \lfloor n/2 \rfloor$. Moreover, in the worst case attacks from $b$ Byzantines, the output $y_i$ and the computational result $c_i$ of node $i$ satisfy $n_0 := |\{i : y_i = 1\}| = |\{i : c_i = 1\}| + b$. Since the estimate on the majority opinion is $\hat{\mu} = \mathrm{maj}\{y_1, \cdots, y_n\}$, the sufficient and necessary condition for perfect Byzantine tolerance (i.e., $\hat{\mu} = \mu$) is $n_0 \leq \lfloor n/2 \rfloor$, or equivalently, $|\{i : c_i = 1\}| = \sum_{v=1}^{\lfloor n/2 \rfloor} |S_v(\boldsymbol{m})| \leq \lfloor n/2 \rfloor - b$. $\qquad\square$

Based on Lemma 2, we can construct explicit matrices $\mathbf{G}$ that guarantee perfect $b-$Byzantine tolerance, under arbitrary $n, b$ settings. The detailed code construction rule is given in Algorithm 1, and the structure of the suggested allocation matrix $\mathbf{G}$ is depicted in Fig. 3. The following theorem states the main property of our code, which is proven in Supplementary Material.

**Theorem 3.** *The deterministic code given in Algorithm 1 satisfies perfect $b-$Byzantine tolerance for $0 < b < \lfloor n/2 \rfloor$, by utilizing a computational redundancy of*

$$ r = \frac{n + (2b + 1)}{2} - \left( \left\lfloor \frac{n - (2b + 1)}{2(b + 1)} \right\rfloor + \frac{1}{2} \right) \frac{n - (2b + 1)}{n}. \tag{5} $$

**Remark 5** (Performance/convergence guarantee). *The code in Algorithm 1 satisfies $\hat{\mu} = \mu$ for any realization of stochastic gradient obtained from $n$ data partitions. Thus, our scheme achieves the ideal performance of SignSGD-MV with no Byzantines, regardless of the actual number of Byzantines in the system. Moreover, the convergence of our scheme is guaranteed from Theorem 2 of [6].*

## 5 Experiments on Amazon EC2

Experimental results are obtained on Amazon EC2. Considering a distributed learning setup, we used message passing interface MPI4py [11].

**Compared schemes.** We compared the suggested coding schemes with the conventional uncoded scheme of SIGNSGD WITH MAJORITY VOTE. Similar to the simulation settings in the previous works [5,6], we used the momentum counterpart SIGNUM instead of SIGNSGD for fast convergence, and used a learning rate of $\gamma = 0.0001$ and a momentum term of $\eta = 0.9$. We simulated deterministic codes given in Algorithm 1, and Bernoulli codes suggested in Section 3 with connection probability of $p$. Thus, the probabilistic code have expected computational redundancy of $\mathbb{E}[r] = np$.

**Byzantine attack model.** We consider two attack models used in related works [6,9]: 1) the *reverse* attack where a Byzantine node flips the sign of the gradient estimate, and 2) the *directional* attack where a Byzantine node guides the model parameter in a certain direction. Here, we set the direction as an all-one vector. For each experiment, $b$ Byzantine nodes are selected arbitrarily.

### 5.1 Experiments on deep neural network models

We trained a RESNET-18 model on CIFAR-10 dataset. Under this setting, the model dimension is $d = 11,173,962$, and the number of training/test samples is set to $n_{\text{train}} = 50000$ and $n_{\text{test}} = 10000$, respectively. We used stochastic mini-batch gradient descent with batch size $B$, and our experiments are simulated on g4dn.xlarge instances (having a GPU) for both workers and the master.

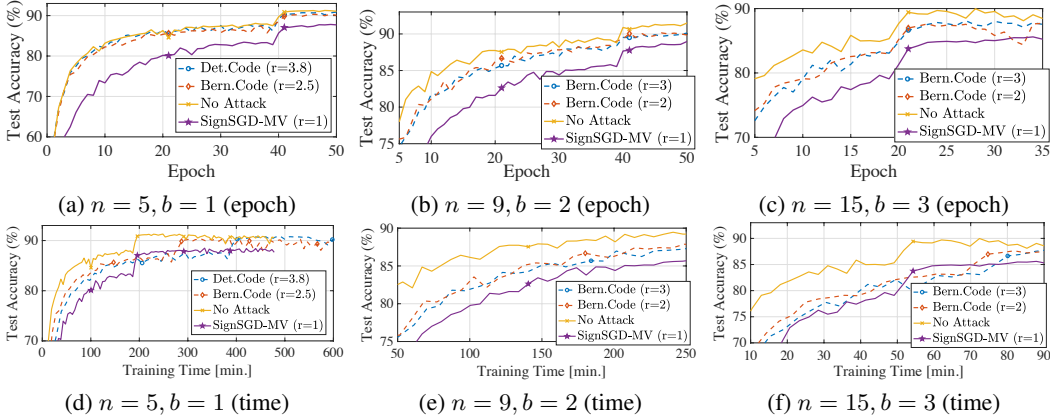

(a) $n = 5, b = 1$ (epoch)   (b) $n = 9, b = 2$ (epoch)   (c) $n = 15, b = 3$ (epoch)

(d) $n = 5, b = 1$ (time)   (e) $n = 9, b = 2$ (time)   (f) $n = 15, b = 3$ (time)

Figure 4: Simulation results with a small portion of Byzantines, under the *reverse* attack on RESNET-18 training CIFAR-10. Deterministic/Bernoulli codes are abbreviated as "Det. code" and "Bern. code", respectively.

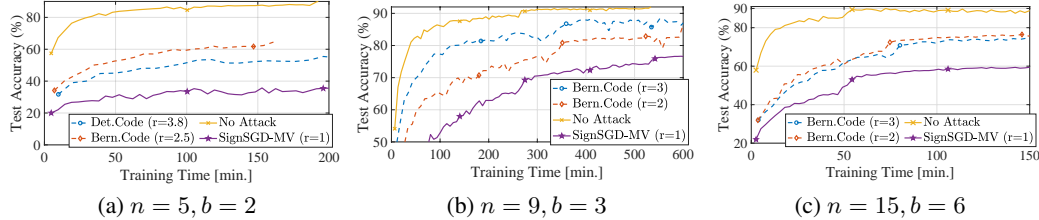

(a) $n = 5, b = 2$   (b) $n = 9, b = 3$   (c) $n = 15, b = 6$

Figure 5: Simulation results with a large portion of Byzantines, under the *reverse* attack on RESNET-18 training CIFAR-10. Deterministic/Bernoulli codes are abbreviated as "Det. code" and "Bern. code", respectively.

Fig. 4 illustrates the test accuracy performances of coded/uncoded schemes, when a small portion of nodes are adversaries. Each curve represents an average over 3 independent train runs. Each column corresponds to different settings of $n$ and $b$: the first scenario is when $n = 5, b = 1$, the second one has $n = 9, b = 2$, and the last setup is $n = 15, b = 3$. For each scenario, two types of plots are given: one having horizontal axis of *training epoch*, and the other with horizontal axis of *training time*. We plotted the case with no attack as a reference to an ideal scenario. As in figures at the top row of Fig. 4, both deterministic and Bernoulli codes nearly achieve the ideal performance at each epoch, for all three scenarios. Overall, the suggested schemes enjoy $5 - 10\%$ accuracy gains compared to the uncoded scheme in [6]. Moreover, the figures at the bottom row of Fig. 4 show that the suggested schemes achieve a given level of test accuracy with less training time, compared to the uncoded scheme. Interestingly, an expected redundancy as small as $\mathbb{E}[r] = 2, 3$ is enough to guarantee a meaningful performance gain compared to the conventional scheme [6] with $r = 1$.

Now, when the Byzantine portion of nodes is large, the results are even more telling. We plotted the simulation results in Fig. 5. Again, each curve reflects an average over 3 independent runs. For various $n, b$ settings, it is clear that the uncoded scheme is highly vulnerable to Byzantine attacks, and the suggested codes with reasonable redundancy (as small as $\mathbb{E}[r] = 2, 3$) help maintaining high accuracy under the attack. When the model converges, the suggested scheme enjoys $15 - 30\%$ accuracy gains compared to the uncoded scheme; our codes provide remarkable training time reductions to achieve a given level of test accuracy.

## 5.2 Experiments on logistic regression models

We trained a logistic regression model on the Amazon Employee Access data set from Kaggle[3], which is used in [26, 30] on coded gradient computation schemes. The model dimension is set to $d = 263500$ after applying one-hot encoding with interaction terms. We used `c4.large` instances for $n$ *workers* that compute batch gradients, and a single `c4.2xlarge` instance for the *master* that aggregates the gradients from workers and determines the model updating rule.

Fig. 6 illustrates the generalization area under curve (AUC) performance of coded/uncoded schemes, under the *directional* attack of Byzantine nodes. Fig. 6a illustrates the performances when $n = 15$ and $b = 5$. Here we set the batch size $B = 15$ and the number of training data $q = 26325$. Fig. 6b compares the performances of coded/uncoded schemes when $n = 49$ and $b = 5$. In this case, we set

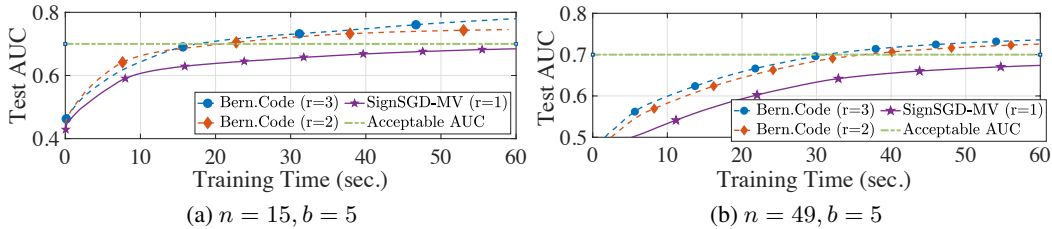

(a) $n = 15, b = 5$         (b) $n = 49, b = 5$

Figure 6: Test AUC performances of coded/uncoded schemes with various redundancy factors $r$, under *directional* attack of $b$ Byzantine nodes. Random Bernoulli codes are abbreviated as "Bern. code".

$B = 5$ and $q = 28665$. In both scenarios, there exist clear performance gaps between the suggested Bernoulli codes and the conventional uncoded scheme. Moreover, our scheme with an expected redundancy as small as $\mathbb{E}[r] = 2$ gives a large training time reduction relative to the uncoded system in achieving a given level of accuracy (e.g. AUC=0.7, regarded as an acceptable performance).

## 6 Conclusions

In this paper, we proposed ELECTION CODING, a coding-theoretic framework that provides Byzantine tolerance of distributed learning employing minimum worker-master communication. This framework tolerates arbitrary attacks corrupting the gradient computed in the training phase, by exploiting redundant gradient computations with appropriate allocation mapping between the individual workers and data partitions. Making use of majority-rule-based encoding as well as decoding functions, we suggested two types of codes that tolerate Byzantine attacks with a controlled amount of redundancy, namely, random Bernoulli codes and deterministic codes. The Byzantine tolerance and the convergence of these coded distributed learning algorithms are proved via mathematical analysis as well as through deep learning and logistic regression simulations on Amazon EC2.

## Broader Impact

Since our scheme uses minimum communication burden across distributed nodes, it is useful for numerous time-sensitive applications including smart traffic systems and anomaly detection in stock markets. Moreover, our work is beneficial for various safety-critical applications that require the highest level of reliability and robustness, including autonomous driving, smart home systems, and healthcare services.

In general, the failure in robustifying machine learning systems may cause some serious problems including traffic accidents or identity fraud. Fortunately, the robustness of the suggested scheme is mathematically proved, so that applying our scheme in safety-critical applications would not lead to such problems.

## Acknowledgments

This work was supported by IITP funds from MSIT of Korea (No. 2016-0-00563 and No. 2020-0-00626) and by National Research Foundation of Korea (No. 2019R1I1A2A02061135 and No. 2020R1A6A3A03039520).

## Footnotes

[1]We assume in this work that Byzantine attacks take place in the communication links and that the distributed nodes themselves are authenticated and trustful. As such, *Byzantine* nodes in our context are the nodes whose transmitted messages are compromised.

[2]According to the assumption on the distribution of $\tilde{g}_j$, no codes satisfy $\mathbb{P}(\hat{\mu} \neq \text{sign}(g)) = 0$. Thus, we instead aim at designing codes that always satisfy $\hat{\mu} = \mu$; when such codes are applied to the system, the learning algorithm is guaranteed to achieve the ideal performance of SignSGD-MV [6] with no Byzantines.

[3]https://www.kaggle.com/c/amazon-employee-access-challenge

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
