[Supplementary Material]

# Election Coding for Distributed Learning: Protecting SignSGD against Byzantine Attacks (Supplementary Materials)

**Jy-yong Sohn**
jysohn1108@kaist.ac.kr

**Dong-Jun Han**
djhan93@kaist.ac.kr

**Beongjun Choi**
bbzang10@kaist.ac.kr

**Jaekyun Moon**
jmoon@kaist.edu

School of Electrical Engineering,
Korea Advanced Institute of Science and Technology (KAIST)

## A  Additional experimental results

### A.1  Speedup of ELECTION CODING over SignSGD-MV

We observe the speedup of ELECTION CODING compared with the conventional uncoded scheme [6]. Table A.1 summarizes the required training time to achieve the target test accuracy, for various $n, b, r$ settings, where $n$ is the number of nodes, $b$ is the number of Byzantines, and $r$ is the computational redundancy. As the portion of Byzantines $\frac{b}{n}$ increases, the suggested code with $r > 1$ has a much higher gain in speedup, compared to the existing uncoded scheme with $r = 1$. Note that this significant gain is achievable by applying codes with a reasonable redundancy of $r = 2, 3$. As an example, our Bernoulli code with mean redundancy of $r = 2$ (two partitions per worker) achieves an 88% accuracy at unit time of 609 while uncoded SignSGD-MV reaches a 79% accuracy at a slower time of 750 and fails altogether to reach the 88% level with $b = 3$ out of $n = 9$ workers compromised.

### A.2  Performances for Byzantine mismatch scenario

Suppose the deterministic code suggested in this paper is designed for $\hat{b} < b$ Byzantine nodes, where $b$ is the actual number of Byzantines in the system. When the number of nodes is $n = 5$ and the number of Byzantines is $b = 2$, Fig. A.1 shows the performance of the deteministic code (with redundancy $r = 3.8$) designed for $\hat{b} = 1$. Even in this underestimated Byzantine setup, the suggested code maintains its tolerance to Byzantine attacks, and the performance gap between the suggested code and the conventional SignSGD-MV is over 20%.

### A.3  Performances for extreme Byzantine attack scenario

In Fig. A.2, we compare the performances of ELECTION CODING and the conventional SIGNSGD-MV, under the maximum number of Byzantine nodes, i.e., $b = (n − 1)/2$, when $n = 9$ and $b = 4$. The suggested Bernoulli code enjoy over 20% performance gap compared to the conventional SignSGD-MV. This shows that the suggested ELECTION CODING is highly robust to Byzantine attacks even under the maximum Byzantine setup, while the conventional SignSGD-MV is vulnerable to the extreme Byzantine attack scenario.

Table A.1: Training time (minutes) to reach the test accuracy for the suggested ELECTION CODING with $r > 1$ and the uncoded SignSGD-MV with $r = 1$, for experiments using ResNet-18 to train CIFAR-10. Here, $\infty$ means that it is impossible for the uncoded scheme to reach the target test accuracy. For every target test accuracy, the suggested code with $r > 1$ requires less training time than the conventional uncoded scheme with $r = 1$.

| Test accuracy | | | 75% | 80% | 85% | 89% |
|---|---|---|---|---|---|---|
| n=5, b=1 | Suggested Election Coding | r=3.8 | **39.2** | **68.6** | **137.2** | **392.1** |
| | | r=2.5 | **28.0** | **56.0** | **119.0** | **287.0** |
| | SignSGD-MV | r=1 | **52.6** | **95.6** | **191.2** | $\infty$ |
| n=9, b=2 | Suggested Election Coding | r=3 | **52.6** | **87.7** | **149.0** | **350.6** |
| | | r=2 | **51.4** | **68.6** | **137.2** | **334.4** |
| | SignSGD-MV | r=1 | **67.0** | **113.8** | **207.5** | **381.5** |

| Test accuracy | | | 30% | 60% | 79% | 88% |
|---|---|---|---|---|---|---|
| n=5, b=2 | Suggested Election Coding | r=3.8 | **9.8** | **19.6** | **58.8** | **245.0** |
| | | r=2.5 | **7.0** | **14.0** | **56.0** | **245.0** |
| | SignSGD-MV | r=1 | **43.0** | $\infty$ | $\infty$ | $\infty$ |
| n=9, b=3 | Suggested Election Coding | r=3 | **8.8** | **26.3** | **122.7** | **394.4** |
| | | r=2 | **8.6** | **60.0** | **351.5** | **608.7** |
| | SignSGD-MV | r=1 | **13.4** | **167.3** | **749.5** | $\infty$ |

Figure A.1: Result for Byzantine mismatch scenario, i.e., $\hat{b} \neq b$, when $n = 5$ and $b = 2$. The suggested deterministic code (with $r = 3.8$) is designed for $\hat{b} = 1$, which is smaller than the actual number of Byzantines $b = 2$. Even in this underestimated Byzantine setup, the suggested code is highly tolerant compared to the conventional SignSGD-MV.

## A.4 Reducing the amount of redundant computation

We remark that there is a simple way to lower the computational burden of ELECTION CODING by reducing the mini-batch size to $\rho B$ for some reduction factor $\rho < 1$. In this case, the effective computational redundancy can be represented as $r_{\text{eff}} = \rho r$. Fig. A.3 shows the performance of ELECTION CODING with reduced effective redundancy. For experiments on $n = 5$ or $n = 15$, we tested the Bernoulli code with redundancy $r = 2.5$ and $r = 3$, respectively, and reduced the batch size by the ratio $\rho = 0.5$. This results in the effective redundancy of $r_{\text{eff}} = 1.25$ and $r_{\text{eff}} = 1.5$, respectively, as in Fig. A.3 . Comparing with the SignSGD with effective redundancy $r_{\text{eff}} = 1$, the suggested codes have $20 \sim 30\%$ performance gap. This shows that ELECTION CODING can be used as a highly practical tool for tolerating Byzantines, with effective redundancy well below 2.

## A.5 Performances for larger networks / noisy computations

We ran experiments for a larger network, ResNet-50, as seen in Fig. A.4a. Our scheme with redundancy $r_{\text{eff}} = 1.5$ is still effective here. We also considered the effect of noisy gradient computation on the performance of the suggested scheme. We added independent Gaussian noise to all gradients corresponding to individual partitions before the signs are taken (in addition to Byzantine attacks on local majority signs). The proposed Bernoulli code can tolerate noisy gradient computation

Figure A.2: Results for maximum Byzantine scenario, i.e., $b = (n-1)/2$, when $n = 9$ and $b = 4$, under the *reverse* attack on RESNET-18 training CIFAR-10. Bernoulli codes are abbreviated as "Bern. code".

(a) $n = 5, b = 2$       (b) $n = 15, b = 6$

Figure A.3: Impact of the reduced effective redundancy $r_{\text{eff}}$, under the *reverse* attack on RESNET-18 training CIFAR-10. The suggested Bernoulli codes with effective redundancy $r_{\text{eff}} = 1.25, 1.5$ has a high performance gain compared to the conventional SignSGD-MV.

(a) ResNet-50     (b) Noisy computation     (c) Compare with median

Figure A.4: Experimental results on CIFAR-10 dataset, for $n = 15$ and $b = 6$ (unless stated otherwise)

well, while uncoded signSGD-MV cannot, as shown in Fig. A.4b for added noise with variance $\sigma^2 = 1e^{-8}$.

## A.6 Comparison with median-based approach

We compared our scheme with full gradient + median (FGM). As in Fig. A.4c, our scheme with redundancy as low as $r_{\text{eff}} = 1.5$ outperforms FGM. While FGM requires no computational redundancy, it needs 32x more communication bandwidth than ours while performing worse. For the 20% Byzantine scenario ($b = 3$), FGM performs relatively better, but still falls well below ours. For the severe 40% Byzantine case ($b = 6$), it is harder to achieve near-perfect protection but our schemes clearly perform better.

## B Hyperparmeter setting in experiments for CIFAR-10 dataset

Experiments for CIFAR-10 dataset on Resnet-18 use the hyperparameters summarized in Table B.2. For the experiments on Resnet-50, the batch size is set to $B = 64$.

Table B.2: Hyperparameters used in experiments for CIFAR-10 dataset on Resnet-18

| $(n, b)$ | (5,1) | (5,2) | (9,2) | (9,3) | (9,4) | (15,3) | (15,6) | (15,7) |
|---|---|---|---|---|---|---|---|---|
| Batch size $B$ (per data partition) | 24 | 48 | 64 | 14 | | 16 | | 64 |
| Learning rate decaying epochs $E$ | [40, 80] | | | | | [20, 40, 80] | | |
| Initial learning rate $\gamma$ | $10^{-4}$ | | | | | | | |
| Momentum $\eta$ | 0.9 | | | | | | | |

## C Notations and preliminaries

We define notations used for proving main mathematical results. For a given set $S$, the identification function $\mathbb{1}_{\{x \in S\}}$ outputs one if $x \in S$, and outputs zero otherwise. We denote the mapping between a message vector $\boldsymbol{m}$ and a coded vector $\boldsymbol{c}$ as $\phi(\cdot)$:

$$\phi(\boldsymbol{m}) = \boldsymbol{c} = [c_1, \cdots, c_n] = [E_1(\boldsymbol{m}; \mathbf{G}), \cdots, E_n(\boldsymbol{m}; \mathbf{G})].$$

We also define the attack vector $\boldsymbol{\beta} = [\beta_1, \beta_2, \cdots, \beta_n]$, where $\beta_j = 1$ if node $j$ is a Byzantine and $\beta_j = 0$ otherwise. The set of attack vectors with a given support $b$ is denoted as $B_b = \{\boldsymbol{\beta} \in \{0,1\}^n :$

Figure C.5: Mapping from $\boldsymbol{m} \in \{0,1\}^n$ to $\hat{\mu} \in \{0,1\}$. For all attack vectors $\boldsymbol{\beta} \in B_b$ and attack functions $f_{\boldsymbol{\beta}} \in \mathcal{F}_{\boldsymbol{\beta}}$, we want the overall mapping to satisfy $\hat{\mu} = 1$ for all $\boldsymbol{m} \in M^-$ and $\hat{\mu} = 0$ for all $\boldsymbol{m} \in M^+$.

$\|\boldsymbol{\beta}\|_0 = b\}$. For a given attack vector $\boldsymbol{\beta}$, we define an attack function $f_{\boldsymbol{\beta}} : \boldsymbol{c} \mapsto \boldsymbol{y}$ to represent the behavior of Byzantine nodes. According to the definition of $y_j$ in the main manuscript, the set of valid attack functions can be expressed as $\mathcal{F}_{\boldsymbol{\beta}} := \{f_{\boldsymbol{\beta}} \in \mathcal{F} : y_j = c_j \quad \forall j \in [n] \text{ with } \beta_j = 0\}$, where $\mathcal{F} = \{f : \{0,1\}^n \to \{0,1\}^n\}$ is the set of all possible mappings. Moreover, the set of message vectors $\boldsymbol{m}$ with weight $t$ is defined as

$$M_t := \{\boldsymbol{m} \in \{0,1\}^n : \|\boldsymbol{m}\|_0 = t\}. \tag{C.1}$$

Now we define several sets:

$$M^+ := \left\{\boldsymbol{m} \in \{0,1\}^n : \|\boldsymbol{m}\|_0 > \left\lfloor \frac{n}{2} \right\rfloor\right\}, \qquad M^- := \left\{\boldsymbol{m} \in \{0,1\}^n : \|\boldsymbol{m}\|_0 \le \left\lfloor \frac{n}{2} \right\rfloor\right\},$$

$$Y^+ := \left\{\boldsymbol{y} \in \{0,1\}^n : \|\boldsymbol{y}\|_0 > \left\lfloor \frac{n}{2} \right\rfloor\right\}, \qquad Y^- := \left\{\boldsymbol{y} \in \{0,1\}^n : \|\boldsymbol{y}\|_0 \le \left\lfloor \frac{n}{2} \right\rfloor\right\}.$$

Using these definitions, Fig. C.5 provides a description on the mapping from $\boldsymbol{m}$ to $\hat{\mu}$. Since decoder $D(\cdot)$ is a majority vote function, we have $\hat{\mu} = \mathbb{1}_{\{\boldsymbol{y} \in Y^+\}}$. Moreover, we have $\mu = \mathbb{1}_{\{\boldsymbol{m} \in M^+\}}$.

Before starting the proofs, we state several preliminary results. We begin with a property, which can be directly obtained from the definition of $\boldsymbol{y} = [y_1, \cdots, y_n]$.

**Lemma C.1.** *Assume that there are $b$ Byzantine nodes, i.e., the attack vector satisfies $\boldsymbol{\beta} \in B_b$. For a given vector $\boldsymbol{c}$, the output $\boldsymbol{y} = f_{\boldsymbol{\beta}}(\boldsymbol{c})$ of an arbitrary attack function $f_{\boldsymbol{\beta}} \in \mathcal{F}_{\boldsymbol{\beta}}$ satisfies $\|\boldsymbol{y} \oplus \boldsymbol{c}\|_0 \le b$. In other words, $\boldsymbol{y}$ and $\boldsymbol{c}$ differ at most $b$ positions.*

Now we state four mathematical results which are useful for proving the theorems in this paper.

**Lemma C.2.** *Consider $X = \sum_{i=1}^n X_i$ where $\{X_i\}_{i \in [n]}$ is the set of independent random variables. Then, the probability density function of $X$ is*

$$f_X = f_{X_1} * \cdots * f_{X_n} = \text{conv}\,\{f_{X_i}\}_{i \in [n]}.$$

**Lemma C.3** (Theorem 2.1, [23])**.** *Consider $f$ and $g$, two arbitrary unimodal distributions that are symmetric around zero. Then, the convolution $f * g$ is also a symmetric unimodal distribution with zero mean.*

**Lemma C.4** (Lemma D.1, [5])**.** *Let $\tilde{g}_k$ be an unbiased stochastic approximation to the $k^{th}$ gradient component $g_k$, with variance bounded by $\sigma_k^2$. Further assume that the noise distribution is unimodal and symmetric. Define the signal-to-noise ratio $S_k := \frac{|g_k|}{\sigma_k}$. Then we have*

$$\mathbb{P}\left[\text{sign}\,(\tilde{g}_k) \ne \text{sign}\,(g_k)\right] \le \begin{cases} \frac{2}{9}\frac{1}{S_k^2} & \text{if } S_i > \frac{2}{\sqrt{3}} \\ \frac{1}{2} - \frac{S_k}{2\sqrt{3}} & \text{otherwise} \end{cases}$$

*which is in all cases less than or equal to $\frac{1}{2}$.*

**Lemma C.5** (Hoeffding's inequality for Binomial distribution)**.** *Let $X = \sum_{i=1}^n X_i$ be the sum of i.i.d. Bernoulli random variables $X_i \sim \text{Bern}(p)$. For arbitrary $\varepsilon > 0$, the tail probability of $X$ is bounded as*

$$\mathbb{P}(X - np \ge n\varepsilon) \le e^{-2\varepsilon^2 n},$$

$$\mathbb{P}(X - np \le -n\varepsilon) \le e^{-2\varepsilon^2 n}.$$

*As an example, when $\varepsilon = \sqrt{\frac{\log n}{n}}$, the upper bound is $\frac{2}{n^2}$, which vanishes as $n$ increases.*

# D Proof of Theorems

## D.1 Proof of Theorem 1

Let $q_{\max} = \max_k q_k$, where $q_k$ is defined in (2). Moreover, define $u_k = 1 - q_k$ and $u_{\min} = 1 - q_{\max}$. Then, $u_{\min}^\star = 1 - \max_k q_k^\star \leq 1 - \max_k q_k = u_{\min}$ holds. Now, we find the global estimation error probability $\mathbb{P}(\hat{\mu}_k \neq \mathrm{sign}(g_k))$ for arbitrary $k$ as below. In the worst case scenario that maximizes the global error, a Byzantine node $i$ always sends the wrong sign bit, i.e., $y_{i,k} \neq \mathrm{sign}(g_k)$. Let $X_{i,k}$ be the random variable which represents whether the $i^{\text{th}}$ Byzantine-free node transmits the correct sign for coordinate $k$:

$$X_{i,k} = \begin{cases} 1, & c_{i,k} = \mathrm{sign}(g_k) \\ 0, & \text{otherwise} \end{cases}$$

Then, $X_{i,k} \sim \mathrm{Bern}(p_k)$. Thus, the number of nodes sending the correct sign bit is $X_{\mathrm{global,k}} = \sum_{i=1}^{n(1-\alpha)} X_{i,k}$, the sum of $X_{i,k}$'s for $n - b = n(1-\alpha)$ Byzantine-free nodes, which follows the binomial distribution $X_{\mathrm{global,k}} \sim \mathcal{B}(n(1-\alpha), p_k)$. From the Hoeffding's inequality (Lemma C.5), we have

$$\mathbb{P}(X_{\mathrm{global,k}} - n(1-\alpha)u_k \leq -n(1-\alpha)\varepsilon_k) \leq e^{-2\varepsilon_k^2 n(1-\alpha)} \tag{D.1}$$

for arbitrary $\varepsilon_k > 0$. Set $\varepsilon_k = u_k - \frac{1}{2(1-\alpha)}$ and define $\varepsilon_{\min} = \min_k \varepsilon_k$. Then, we have

$$u_{\min}^\star - \frac{1}{2(1-\alpha)} > \sqrt{\frac{1}{2(1-\alpha)}}\sqrt{\frac{\log(\Delta)}{n}}, \tag{D.2}$$

which is proven as below. Let $Y = \sqrt{2(1-\alpha)}$. Then, (D.2) is equivalent to

$$u_{\min}^\star Y^2 - \sqrt{\frac{\log(\Delta)}{n}}Y - 1 > 0, \tag{D.3}$$

which is all we need to prove. Note that (D.2) implies that

$$\frac{Y^2}{2} > \frac{(\sqrt{\log(\Delta)/n} + \sqrt{\log(\Delta)/n + 4u_{\min}^\star})^2}{8(u_{\min}^\star)^2},$$

which is equivalent to

$$Y > \frac{\sqrt{\log(\Delta)/n}}{2u_{\min}^\star} + \frac{\sqrt{\log(\Delta)/n + 4u_{\min}^\star}}{2u_{\min}^\star}. \tag{D.4}$$

Then,

$$u_{\min}^\star Y^2 - \sqrt{\frac{\log(\Delta)}{n}}Y - 1 = u_{\min}^\star\left(Y - \frac{1}{2u_{\min}^\star}\sqrt{\frac{\log(\Delta)}{n}}\right)^2 - \frac{1}{4u_{\min}^\star}\frac{\log(\Delta)}{n} - 1 > 0,$$

which proves (D.3) and thus (D.2). Thus, we have $\varepsilon_k \geq \varepsilon_{\min} \geq u_{\min}^\star - \frac{1}{2(1-\alpha)} > \sqrt{\frac{1}{2(1-\alpha)}}\sqrt{\frac{\log(\Delta)}{n}} > 0$. Then, (D.1) reduces to

$$P_{\mathrm{global,k}} = \mathbb{P}(\hat{\mu}_k \neq \mathrm{sign}(g_k)) = \mathbb{P}(X_{\mathrm{global,k}} \leq n/2) \leq e^{-2\varepsilon_k^2 n(1-\alpha)} < 1/\Delta,$$

which completes the proof.

## D.2 Proof of Theorem 2

Here we basically follow the proof of [6] with a slight modification, reflecting the result of Theorem 1. Let $\boldsymbol{w}_1, \cdots, \boldsymbol{w}_T$ be the sequence of updated models at each step. Then, we have

$$f(\boldsymbol{w}_{t+1}) - f(\boldsymbol{w}_t) \overset{(a)}{\leq} g(\boldsymbol{w}_t)^T(\boldsymbol{w}_{t+1} - \boldsymbol{w}_t) + \sum_{k=1}^d \frac{L_k}{2}(\boldsymbol{w}_{t+1} - \boldsymbol{w}_t)_k^2 \overset{(b)}{=} -\gamma g(\boldsymbol{w}_t)^T \hat{\boldsymbol{\mu}} + \gamma^2 \sum_{k=1}^d \frac{L_k}{2}$$

$$= -\gamma\|g(\boldsymbol{w}_t)\|_1 + 2\gamma \sum_{i=1}^d |(g(\boldsymbol{w}_t))_k| \cdot \mathbb{1}_{\{\mathrm{sign}((g(\boldsymbol{w}_t))_k) \neq \hat{\mu}_k\}} + \gamma^2 \sum_{k=1}^d \frac{L_k}{2},$$

where (a) is from Assumption 2, and (b) is obtained from $\boldsymbol{w}_{t+1} = \boldsymbol{w}_t - \gamma\hat{\boldsymbol{\mu}}$ and $|\hat{\mu}_k| = 1$. Let $g_k$ be a simplified notation of $(g(\boldsymbol{w}_t))_k$. Then, taking the expectation of the equation above, we have

$$\mathbb{E}\left[f(\boldsymbol{w}_{t+1}) - f(\boldsymbol{w}_t)\right] \leq -\gamma\|g(\boldsymbol{w}_t)\|_1 + \frac{\gamma^2}{2}\|\boldsymbol{L}\|_1 + 2\gamma\sum_{k=1}^d |g_k|\,\mathbb{P}(\hat{\mu}_k \neq \text{sign}(g_k)). \quad (D.5)$$

Inserting the result of Theorem 1 to (D.5), we have

$$\mathbb{E}\left[f(\boldsymbol{w}_{t+1}) - f(\boldsymbol{w}_t)\right] \leq -\gamma\left(1 - \frac{2}{\Delta}\right)\|g(\boldsymbol{w}_t)\|_1 + \frac{\gamma^2}{2}\|\boldsymbol{L}\|_1$$

$$\overset{(a)}{=} -\sqrt{\frac{f(\boldsymbol{w}_0) - f^*}{\|\boldsymbol{L}\|_1 T}}\left(1 - \frac{2}{\Delta}\right)\|g(\boldsymbol{w}_t)\|_1 + \frac{f(\boldsymbol{w}_0) - f^*}{2T}$$

where (a) is from the parameter settings of $\gamma$ in the statement of Theorem 2. Thus, we have

$$f(\boldsymbol{w}_0) - f^* \geq f(\boldsymbol{w}_0) - \mathbb{E}\left[f(\boldsymbol{w}_T)\right] = \sum_{t=0}^{T-1}\mathbb{E}\left[f(\boldsymbol{w}_t) - f(\boldsymbol{w}_{t+1})\right]$$

$$\geq \sum_{t=0}^{T-1}\mathbb{E}\left[\sqrt{\frac{f(\boldsymbol{w}_0) - f^*}{\|\boldsymbol{L}\|_1 T}}\left(1 - \frac{2}{\Delta}\right)\|g(\boldsymbol{w}_t)\|_1 - \frac{f(\boldsymbol{w}_0) - f^*}{2T}\right]$$

$$= \sqrt{\frac{T\left(f(\boldsymbol{w}_0) - f^*\right)}{\|\boldsymbol{L}\|_1}}\,\mathbb{E}\left[\frac{1}{T}\sum_{t=0}^{T-1}\|g(\boldsymbol{w}_t)\|_1\right]\left(1 - \frac{2}{\Delta}\right) - \frac{f(\boldsymbol{w}_0) - f^*}{2}.$$

The expected gradient (averaged out over $T$ iterations) is expressed as

$$\frac{1}{T}\sum_{t=0}^{T-1}\mathbb{E}\left[\|g(\boldsymbol{w}_t)\|_1\right] \leq \frac{1}{\sqrt{T}}\frac{1}{1 - \frac{2}{\Delta}}\frac{3}{2}\sqrt{\|\boldsymbol{L}\|_1(f(\boldsymbol{w}_0) - f^*)} \to 0 \quad \text{as } T \to \infty$$

Thus, the gradient becomes zero eventually, which completes the convergence proof.

### D.3 Proof of Theorem 3

Recall that according to Lemma 2 and the definition of $M_t$ in (C.1), the system using the allocation matrix $\mathbf{G}$ is perfect $b$−Byzantine tolerable if and only if

$$\sum_{v=1}^{(n-1)/2}|S_v(\boldsymbol{m})| \leq \frac{n-1}{2} - b \quad (D.6)$$

holds for arbitrary message vector $\boldsymbol{m} \in M_{\frac{n-1}{2}}$, where $S_v(\boldsymbol{m})$ is defined in (4). Note that we have

$$\|\mathbf{G}(j,:)\|_0 = \begin{cases} 1, & 1 \leq j \leq s \\ 2b+1, & s+1 \leq j \leq s+L \\ n, & s+L+1 \leq j \leq n. \end{cases} \quad (D.7)$$

from Fig. 3. Thus, the condition in (D.6) reduces to

$$|S_1(\boldsymbol{m})| + |S_{b+1}(\boldsymbol{m})| \leq s. \quad (D.8)$$

Now all that remains is to show that (D.8) holds for arbitrary message vector $\boldsymbol{m} \in M_{\frac{n-1}{2}}$.

Consider a message vector $\boldsymbol{m} \in M_{\frac{n-1}{2}}$ denoted as $\boldsymbol{m} = [m_1, m_2, \cdots, m_n]$. Here, we note that

$$S_1(\boldsymbol{m}) \subseteq \{1, 2, \cdots, s\}, \quad S_{b+1}(\boldsymbol{m}) \subseteq \{s+1, s+2, \cdots, s+L\} \quad (D.9)$$

hold from Fig. 3. Define

$$v(\boldsymbol{m}) := |\{i \in \{s+1, s+2, \cdots, n\} : m_i = 1\}|, \quad (D.10)$$

Figure D.6: Sets of message vectors used in proving Lemmas D.1 and D.2

which is the number of 1's in the last $(n - s)$ coordinates of message vector $\boldsymbol{m}$. Since $\boldsymbol{m} \in M_{\frac{n-1}{2}}$, we have

$$|\{i \in \{1, 2, \cdots, s\} : m_i = 1\}| = \frac{n-1}{2} - v(\boldsymbol{m}). \tag{D.11}$$

Note that since $\mathbf{G}(1 : s, :) = [\, \mathbf{I}_s \mid \mathbf{0}_{s \times (n-s)}]$, we have

$$\boldsymbol{m}^T \mathbf{G}(j, :) = \mathbb{1}_{\{m_j = 1\}}, \quad \|\mathbf{G}(j, :)\|_0 = 1, \quad \forall j \in [s]. \tag{D.12}$$

Combining (4), (D.9), (D.11), and (D.12), we have $|S_1(\boldsymbol{m})| = \frac{n-1}{2} - v(\boldsymbol{m})$. Now, in order to obtain (D.8), all we need to prove is to show

$$|S_{b+1}(\boldsymbol{m})| \leq s - \left( \frac{n-1}{2} - v(\boldsymbol{m}) \right) \overset{(a)}{=} v(\boldsymbol{m}) - b \tag{D.13}$$

where $(a)$ is from the definition of $s$ in Algorithm 1. We alternatively prove that[4]

$$\text{if } |S_{b+1}(\boldsymbol{m})| \geq q \text{ for some } q \in \{0, 1, \cdots, L\}, \text{ then } v(\boldsymbol{m}) \geq b + q. \tag{D.14}$$

Using the definition $M^{(q)} := \{\boldsymbol{m} \in M_{\frac{n-1}{2}} : |S_{b+1}(\boldsymbol{m})| \geq q\}$, the statement in (D.14) is proved as follows: for arbitrary $q \in \{0, 1, \cdots, L\}$, we first find the minimum $v(\boldsymbol{m})$ among $\boldsymbol{m} \in M^{(q)}$, i.e., we obtain a closed-form expression for

$$v_q^* := \min_{\boldsymbol{m} \in M^{(q)}} v(\boldsymbol{m}). \tag{D.15}$$

Second, we show that $v_q^* \geq b + q$ holds for all $q \in \{0, 1, \cdots, L\}$, which completes the proof.

The expression for $v_q^*$ can be obtained as follows. Fig. D.6 supports the explanation. First, define

$$M_{gather}^{(q)} := \{\boldsymbol{m} \in M^{(q)} : \text{ if } j, j + 2 \in S_{b+1}(\boldsymbol{m}), \text{ then } j + 1 \in S_{b+1}(\boldsymbol{m})\}, \tag{D.16}$$

the set of message vectors $\boldsymbol{m}$ which satisfy that $S_{b+1}(\boldsymbol{m})$ is consisted of consecutive integers. We now provide a lemma which states that within $M_{gather}^{(q)}$, there exists a minimizer of the optimization problem (D.15).

**Lemma D.1.** *For arbitrary $q \in \{0, 1, \cdots, L\}$, we have*

$$v_q^* = \min_{\boldsymbol{m} \in M_{gather}^{(q)}} v(\boldsymbol{m}).$$

*Proof.* From Fig. D.6 and the definition of $v_q^*$, all we need to prove is the following statement: for all $\boldsymbol{m} \in M^{(q)} \cap (M_{gather}^{(q)})^c$, we can assign another message vector $\boldsymbol{m}^* \in M_{gather}^{(q)}$ such that $v(\boldsymbol{m}^*) \leq v(\boldsymbol{m})$ holds. Consider arbitrary $\boldsymbol{m} \in M^{(q)} \cap (M_{gather}^{(q)})^c$, denoted as $\boldsymbol{m} = [m_1, m_2, \cdots, m_n]$. Then, there exist integers $j \in \{1, \cdots, L\}$ and $\delta \in \{2, 3, \cdots, L - j\}$ such that $s + j, s + j + \delta \in S_{b+1}(\boldsymbol{m})$ and $s + j + 1, \cdots, s + j + \delta - 1 \notin S_{b+1}(\boldsymbol{m})$ hold. Select the smallest $j$ which satisfies the condition. Consider $\boldsymbol{m}' = [m_1', \cdots, m_n']$ generated as the following rule:

1. The first $s + j(b + 1)$ elements (which affect the first $j$ rows of $\mathbf{A}$ in Figure 3) of $\boldsymbol{m}'$ is identical to that of $\boldsymbol{m}$.

2. The last $n - (j + \delta - 1)(b+1) - s$ elements of $\boldsymbol{m}$ are shifted to the left by $(\delta - 1)(b+1)$, and inserted to $\boldsymbol{m}'$. In the shifting process, we have $b$ locations where the original $m_i$ and the shifted $m_{i+(\delta-1)(b+1)}$ overlap. In such locations, $m_i'$ is set to the maximum of two elements; if either one is 1, we set $m_i' = 1$, and otherwise we set $m_i' = 0$.

This can be mathematically expressed as below:

$$m_i' = \begin{cases} m_i, & 1 \leq i \leq s + j(b+1) \\ \max\{m_i, m_{i+(\delta-1)(b+1)}\}, & s + j(b+1) + 1 \leq i \leq s + (j+1)(b+1) \\ m_{i+(\delta-1)(b+1)}, & s + (j+1)(b+1) + 1 \leq i \leq n - (\delta-1)(b+1) \\ 0, & n - (\delta-1)(b+1) + 1 \leq i \leq n \end{cases} \tag{D.17}$$

Note that we have

$$\sum_{i=1}^{n} m_i = \frac{n-1}{2} \tag{D.18}$$

since $\boldsymbol{m} \in M_{(n-1)/2}$. Moreover, (D.17) implies

$$\sum_{i=1}^{n} m_i' = \sum_{i=1}^{s+j(b+1)} m_i' + \sum_{i=s+j(b+1)+1}^{s+(j+1)(b+1)} m_i' + \sum_{i=s+(j+1)(b+1)+1}^{n-(\delta-1)(b+1)} m_i'$$

$$= \sum_{i=1}^{s+j(b+1)} m_i + \sum_{i=s+j(b+1)+1}^{s+(j+1)(b+1)} m_i' + \sum_{i=s+(j+\delta)(b+1)+1}^{n} m_i$$

$$\overset{\text{(D.18)}}{=} \frac{n-1}{2} - \left( \sum_{i=s+j(b+1)+1}^{s+(j+\delta)(b+1)} m_i - \sum_{i=s+j(b+1)+1}^{s+(j+1)(b+1)} m_i' \right) \overset{\text{(a)}}{\leq} \frac{n-1}{2} \tag{D.19}$$

where Eq.$(a)$ is from

$$\sum_{i=s+j(b+1)+1}^{s+(j+\delta)(b+1)} m_i \overset{\text{(b)}}{\geq} \sum_{i=s+j(b+1)+1}^{s+(j+1)(b+1)} (m_i + m_{i+(\delta-1)(b+1)}) \geq \sum_{i=s+j(b+1)+1}^{s+(j+1)(b+1)} \max\{m_i, m_{i+(\delta-1)(b+1)}\}$$

$$\overset{\text{(D.17)}}{=} \sum_{i=s+j(b+1)+1}^{s+(j+1)(b+1)} m_i'$$

and Eq.$(b)$ is from $\delta \geq 2$. Note that

$$v(\boldsymbol{m}') = v(\boldsymbol{m}) - \varepsilon \tag{D.20}$$

holds for

$$\varepsilon := \frac{n-1}{2} - \sum_{i=1}^{n} m_i', \tag{D.21}$$

which is a non-negative integer from (D.19). Now, we show the behavior of $S_{b+1}(\boldsymbol{m})$ as follows. Recall that for $j_0 \in \{s+1, \cdots, s+L\}$,

$$\mathbf{G}(j_0, i_0) = \begin{cases} 1, & \text{if } s + (j_0 - s - 1)(b+1) + 1 \leq i_0 \leq s + (j_0 - s - 1)(b+1) + 2b + 1 \\ 0, & \text{otherwise} \end{cases} \tag{D.22}$$

holds from Algorithm 1 and Fig. 3. Define

$$S_+ := \{j' \in \{s+j+\delta, \cdots, s+L\} : j' \in S_{b+1}(\boldsymbol{m})\},$$
$$S_- := \{j' \in \{s+1, \cdots, s+j\} : j' \in S_{b+1}(\boldsymbol{m})\}.$$

From (D.17) and (D.22), we have

$$\begin{cases} S_- \subseteq S_{b+1}(\boldsymbol{m}'), \\ \text{if } j' \in S_+, \text{ then } j' - (\delta - 1) \in S_{b+1}(\boldsymbol{m}'). \end{cases}$$

Thus, we have

$$|S_{b+1}(\boldsymbol{m}')| \geq |S_-| + |S_+| = |S_{b+1}(\boldsymbol{m})| \overset{(a)}{\geq} q \qquad \text{(D.23)}$$

where Eq.$(a)$ is from $\boldsymbol{m} \in M^{(q)}$.

Now we construct $\boldsymbol{m}'' \in M^{(q)}$ which satisfies $v(\boldsymbol{m}'') \leq v(\boldsymbol{m})$. Define $S_0 := \{i \in [s] : m_i' = 0\}$ and $\varepsilon_0 := |S_0|$. The message vector $\boldsymbol{m}'' = [m_1'', \cdots, m_n'']$ is defined as follows.

**Case I** (when $\varepsilon \leq \varepsilon_0$): Set $m_i'' = m_i'$ for $i \in \{s+1, s+2, \cdots, n\}$. Randomly select $\varepsilon$ elements in $S_0$, denoted as $\{i_1, \cdots, i_\varepsilon\} = S_0^{(\varepsilon)} \subseteq S_0$. Set $m_i'' = 1$ for $i \in S_0^{(\varepsilon)}$, and set $m_i'' = m_i'$ for $i \in S_0 \setminus S_0^{(\varepsilon)}$. Note that this results in

$$v(\boldsymbol{m}'') = v(\boldsymbol{m}'). \qquad \text{(D.24)}$$

**Case II** (when $\varepsilon > \varepsilon_0$): Set $m_i'' = 1$ for $i \in [s]$. Define $S_1 := \{i \in \{s+1, \cdots, n\} : m_i' = 0\}$. Randomly select $\varepsilon - \varepsilon_0$ elements in $S_1$, denoted as $\{i_1', \cdots, i_{\varepsilon-\varepsilon_0}'\} = S_1^{(\varepsilon)} \subseteq S_1$. Set $m_i'' = 1$ for $i \in S_1^{(\varepsilon)}$, and set $m_i'' = m_i'$ for $i \in \{s+1, \cdots, n\} \setminus S_1^{(\varepsilon)}$. Note that this results in

$$v(\boldsymbol{m}'') = v(\boldsymbol{m}') + (\varepsilon - \varepsilon_0). \qquad \text{(D.25)}$$

Note that in both cases, the weight of $\boldsymbol{m}''$ is

$$\|\boldsymbol{m}''\|_0 = \sum_{i=1}^n m_i'' = \sum_{i=1}^n m_i' + \varepsilon \overset{\text{(D.21)}}{=} \frac{n-1}{2}. \qquad \text{(D.26)}$$

Moreover,

$$|S_{b+1}(\boldsymbol{m}'')| \overset{(a)}{\geq} |S_{b+1}(\boldsymbol{m}')| \overset{\text{(D.23)}}{\geq} q \qquad \text{(D.27)}$$

holds where Eq.$(a)$ is from the fact that all elements of $\boldsymbol{m}'' - \boldsymbol{m}$ are non-negative. Finally,

$$v(\boldsymbol{m}'') = v(\boldsymbol{m}) - \min\{\varepsilon, \varepsilon_0\} \leq v(\boldsymbol{m}) \qquad \text{(D.28)}$$

holds from (D.20), (D.24), and (D.25). Combining (D.26), (D.27) and (D.28), one can confirm that $\boldsymbol{m}'' \in M^{(q)}$ and $v(\boldsymbol{m}'') \leq v(\boldsymbol{m})$ hold; this *gathering* process[5] maintains the weight of a message vector and does not increase the $v$ value. Let $\boldsymbol{m}^*$ be the message vector generated by applying this gathering process to $\boldsymbol{m}$ sequentially until $S_{b+1}(\boldsymbol{m}^*)$ is consisted of consecutive integers. Then, $\boldsymbol{m}^*$ satisfies the followings:

1. $S_{b+1}(\boldsymbol{m}^*)$ contains more than $q$ elements. Moreover, since $S_{b+1}(\boldsymbol{m}^*)$ is consisted of consecutive integers, we have $\boldsymbol{m}^* \in M_{gather}^{(q)}$.

2. $v(\boldsymbol{m}^*) \leq v(\boldsymbol{m}'') \leq v(\boldsymbol{m})$ holds.

Since the above process of generating $\boldsymbol{m}^* \in M_{gather}^{(q)}$ is valid for arbitrary message vector $\boldsymbol{m} \in M^{(q)} \cap (M_{gather}^{(q)})^c$, this completes the proof.

Now consider arbitrary message vectors satisfying $\boldsymbol{m} \in M_{gather}^{(q)}$. Then, we have

$$S_{b+1}(\boldsymbol{m}) = \{j, j+1, \cdots, j+\delta-1\} \qquad \text{(D.29)}$$

for some $j \in \{s+1, \cdots, s+L-\delta+1\}$ and $\delta \geq q$. Here, we define

$$M_{gather,overlap}^{(q)} = \left\{ \boldsymbol{m} \in M_{gather}^{(q)} : m_{s+(j_0-s)(b+1)} = 0 \text{ for } j_0 \in \{j, \cdots, j+\delta-1\} \right\} \qquad \text{(D.30)}$$

We here prove that arbitrary message vector $\boldsymbol{m} \in M_{gather}^{(q)}$ can be mapped into another message vector $\boldsymbol{m}' \in M_{gather,overlap}^{(q)}$ without increasing the corresponding $v$ value, i.e., $v(\boldsymbol{m}') \leq v(\boldsymbol{m})$.

Figure D.7: The gathering process illustrated in the proof of Lemma D.1, when $n = 17, b = 2$. Under this setting, we have $S_{b+1}(\boldsymbol{m}) = S_3(\boldsymbol{m}) = \{j \in \{s+1, \cdots, s+L\} : \boldsymbol{m}^T G(j,:) \geq 3\}$. Before the gathering process, we have $S_{b+1}(\boldsymbol{m}) = \{s+1, s+3\}$, while $S_{b+1}(\boldsymbol{m}'') = \{s+1, s+2\}$ holds after the process.

Given a message vector $\boldsymbol{m} \in M_{gather}^{(q)}$, define $\boldsymbol{m}' = [m_1', m_2', \cdots, m_n']$ as in Algorithm 2. In line 9 of this algorithm, we can always find $l \in [s]$ that satisfies $m_l = 0$, due to the following reason. Note that

$$\sum_{i=s+1}^{n} m_i \overset{(a)}{\geq} \boldsymbol{m}^T \mathbf{G}(j,:) \overset{(D.29)}{\geq} b+1 \tag{D.31}$$

holds where $(a)$ is from the fact that $G(j,i) = 0$ for $i \in [s]$ as in (D.22). Thus, we have

$$\sum_{i=1}^{s} m_i \overset{(D.29)}{\leq} \sum_{i=1}^{n} m_i - (b+1) = \frac{n-1}{2} - (b+1) = s-1. \tag{D.32}$$

Therefore, we have

$$\exists l \in [s] \text{ such that } m_l = 0. \tag{D.33}$$

The vector $\boldsymbol{m}'$ generated from Algorithm 2 satisfies the following four properties:

1. $\boldsymbol{m}' \in M_{(n-1)/2}$,

2. $S_{b+1}(\boldsymbol{m}') = S_{b+1}(\boldsymbol{m}) = \{j, j+1, \cdots, j+\delta-1\}$,

3. $\boldsymbol{m}' \in M_{gather,overlap}^{(q)}$,

4. $v(\boldsymbol{m}') \leq v(\boldsymbol{m})$.

The first property is from the fact that lines 7 and 10 of the algorithm maintains the weight of the message vector to be $\|\boldsymbol{m}\|_0 = (n-1)/2$. The second property is from the fact that

$$(\boldsymbol{m}')^T \mathbf{G}(j_0,:) \overset{(D.22)}{=} \sum_{i=1}^{2b+1} m_{s+(j_0-s-1)(b+1)+i}'$$

$$\overset{(a)}{=} \begin{cases} \sum_{i=1}^{2b+1} m_{s+(j_0-s-1)(b+1)+i}, & \text{if line 6 of Algorithm 2 is satisfied} \\ 2b, & \text{otherwise} \end{cases} \overset{(D.29)}{\geq} b+1$$

for $j_0 \in \{j, j+1, \cdots, j+\delta-1\}$, where $(a)$ is from the fact that $\sum_{i=1}^{2b+1} m_{s+(j_0-s-1)(b+1)+i} = 2b+1$ holds if line 6 of Algorithm 2 is not satisfied. The third property is from the first two properties and

**Algorithm 2** Defining $\boldsymbol{m}' \in M_{gather,overlap}^{(q)}$ from arbitrary $\boldsymbol{m} \in M_{gather}^{(q)}$.

---

1: **Input:** message vector $\boldsymbol{m} = [m_1, m_2, \cdots, m_n]$ having $S_{b+1}(\boldsymbol{m}) = \{j, j+1, \cdots, j+\delta-1\}$
2: **Output:** message vector $\boldsymbol{m}' = [m_1', m_2', \cdots, m_n']$
3: **Initialize:** $\boldsymbol{m}' = \boldsymbol{m}$
4: **for** $j_0 = j$ to $j + \delta - 1$ **do**
5:   **if** $m_{s+(j_0-s)(b+1)} = 1$ **then**
6:     **if** $\exists i \in [2b+1]$ such that $m_{s+(j_0-s-1)(b+1)+i} = 0$ **then**
7:       $m_{s+(j_0-s-1)(b+1)+i}' \leftarrow 1, \quad m_{s+(j_0-s)(b+1)}' \leftarrow 0.$
8:     **else**
9:       Find $l \in [s]$ such that $m_l = 0$ (The existence of such $l$ is proven in (D.33).)
10:       $m_l' \leftarrow 1, \quad m_{s+(j_0-s)(b+1)}' \leftarrow 0.$
11:     **end if**
12:   **end if**
13: **end for**

---

Figure D.8: The illustration of overlapping regions and $\{a_l\}_{l=0}^{\delta}$ in (D.35)

the definition of $M_{gather,overlap}^{(q)}$ in (D.30). The last property is from the fact that 1) each execution of line 7 in the algorithm maintains $v(\boldsymbol{m}') = v(\boldsymbol{m})$, and 2) each execution of line 10 in the algorithm results in $v(\boldsymbol{m}') = v(\boldsymbol{m}) - 1$. Thus, combining with Lemma D.1, we have the following lemma:

**Lemma D.2.** *For arbitrary $q \in \{0, 1, \cdots, L\}$, we have*

$$v_q^* = \min_{\boldsymbol{m} \in M_{gather,overlap}^{(q)}} v(\boldsymbol{m}).$$

According to Lemma D.2, in order to find $v_q^*$, all that remains is to find the optimal $\boldsymbol{m} \in M_{gather,overlap}^{(q)}$ which has the minimum $v(\boldsymbol{m})$. Consider arbitrary $\boldsymbol{m} \in M_{gather,overlap}^{(q)}$ and denote

$$S_{b+1}(\boldsymbol{m}) = \{j, j+1, \cdots, j+\delta-1\}. \tag{D.34}$$

Define the corresponding assignment vector $\{a_l\}_{l=0}^{\delta}$ as

$$a_l = \sum_{i=1}^{b} m_{s+(j-s-1+l)(b+1)+i}, \tag{D.35}$$

which represents the number of indices $i$ satisfying $m_i = 1$ within $l^{\text{th}}$ overlapping region, as illustrated in Fig. D.8. Then, we have

$$
\begin{aligned}
a_{j_0-j} + a_{j_0-j+1} &= \sum_{i=1}^{b} m_{s+(j_0-s-1)(b+1)+i} + m_{s+(j_0-s)(b+1)+i} \\
&\overset{(D.30)}{=} \sum_{i=1}^{2b+1} m_{s+(j_0-s-1)(b+1)+i} \overset{(D.22),(D.34)}{\geq} b+1
\end{aligned} \tag{D.36}
$$

for $j_0 \in \{j, j+1, \cdots, j+\delta-1\}$. Since $a_t$ is the sum of $b$ binary elements, we have

$$1 \le a_t \le b, \quad \forall t \in \{0, 1, \cdots, \delta\} \tag{D.37}$$

from (D.36). Now define a message vector $\boldsymbol{m}' \in M^{(q)}_{gather,overlap}$ satisfying the followings: the corresponding assignment vector is

$$(a'_0, a'_1, \cdots, a'_\delta) = \begin{cases} (1, b, 1, b, \cdots, 1, b), & \text{if } \delta \text{ is odd} \\ (1, b, 1, b, \cdots, 1), & \text{otherwise,} \end{cases} \tag{D.38}$$

for $a'_l = \sum_{i=1}^{b} m'_{s+(j-s-1+l)(b+1)+i}$, and the elements $m'_i$ for $i \in [s]$ is $m'_i = \mathbb{1}_{\{i \le i_{max}\}}$ where $i_{max} = \frac{n-1}{2} - \sum_{l=0}^{\delta} a'_l \le s$. Then, we have

$$v(\boldsymbol{m}) \ge \sum_{l=0}^{\delta} a_l \overset{\text{(D.36),(D.37)}}{\ge} \sum_{l=0}^{\delta} a'_l \overset{\text{(D.38)}}{=} v(\boldsymbol{m}') \tag{D.39}$$

for arbitrary $\boldsymbol{m} \in M^{(q)}_{gather,overlap}$. Moreover, among $\delta \ge q$, setting $\delta = q$ minimizes $v(\boldsymbol{m}')$, having the optimum value of

$$v_q^* \overset{\text{(a)}}{=} v(\boldsymbol{m}') \overset{\text{(D.38)}}{=} \begin{cases} \sum_{i=1}^{\frac{q+1}{2}}(1+b), & \text{if } q \text{ is odd} \\ 1 + \sum_{i=1}^{\frac{q}{2}}(1+b), & \text{otherwise} \end{cases} \overset{\text{(b)}}{\ge} \begin{cases} 1 + b + 2(\frac{q+1}{2}-1) = b+q, & \text{if } q \text{ is odd} \\ 1 + (1+b) + 2(\frac{q}{2}-1) = b+q, & \text{otherwise} \end{cases}$$

where $(a)$ is from (D.39) and Lemma D.2, and $(b)$ is from $b \ge 1$. Combining this with the definition of $v_q^*$ in (D.15) proves (D.14). This completes the proofs for (D.8) and (D.6). Thus, the data allocation matrix $\mathbf{G}$ in Algorithm 1 perfectly tolerates $b$ Byzantines. From Fig. 3, the required redundancy of this code is

$$r = \frac{s + (2b+1)L + n(n-s-L)}{n} \overset{\text{(a)}}{=} \frac{n - (2b+1)}{2n} + \frac{2b+1}{n}L + \left(\frac{n+(2b+1)}{2} - L\right)$$

$$= \frac{n+(2b+1)}{2} - \left(L - \frac{1}{2}\right)\frac{n-(2b+1)}{n},$$

where Eq.$(a)$ is from the definition of $s$ in Algorithm 1. $\square$

# E  Proof of Lemmas and Propositions

## E.1  Proof of Lemma 1

We start from finding the estimation error $q_{k|n_i}$ of an arbitrary node $i$ having $n_i$ data partitions.

**Lemma E.1** (conditional local error). *Suppose $n_i$ data partitions are assigned to a Byzantine-free node $i$. Then, the probability of this node transmitting a wrong sign to PS for coordinate $k$ is bounded as*

$$q_{k|n_i} = \mathbb{P}(c_{i,k} \ne \text{sign}(g_k)|n_i) \le \exp\left(-n_i S_k^2 / \{2(S_k^2 + 4)\}\right). \tag{E.1}$$

This lemma is proven by applying Hoeffding's inequality for binomial random variable, as shown in Section E.3 of the Supplementary Materials. The remark below summaries the behavior of the local estimation error as the number of data partitions assigned to a node increases.

**Remark 6.** *The error bound in Lemma E.1 is an exponentially decreasing function of $n_i$. This implies that as the number of data partitions assigned to a node increases, it is getting highly probable that the node correctly estimates the sign of true gradient. This supports the experimental results in Section 4 showing that random Bernoulli codes with a small amount of redundancy (e.g. $\mathbb{E}[r] = 2, 3$) are enough to enjoy a significant gap compared to the conventional SignSGD-MV [6] with $r = 1$.*

Note that $n_i \sim \mathcal{B}(n, p)$ is a binomial random variable. Based on the result of Lemma E.1, we obtain the local estimation error $q_k$ by averaging out $q_{k|n_i}$ over all realizations of $n_i$. Define

$\varepsilon = p^\star/2 = \sqrt{C\log(n)/n}$. Then, we calculate the failure probability of node $i$ as

$$
\begin{aligned}
q_k &= \mathbb{P}(c_{i,k} \neq \text{sign}(g_k)) = \sum_{n_i=0}^{n} \mathbb{P}(n_i)\mathbb{P}(c_{i,k} \neq \text{sign}(g_k) \mid n_i) \\
&= \sum_{n_i:|n_i-np|\geq n\varepsilon} \mathbb{P}(n_i)\mathbb{P}(c_{i,k} \neq \text{sign}(g_k) \mid n_i) + \sum_{n_i:|n_i-np|<n\varepsilon} \mathbb{P}(n_i)\mathbb{P}(c_{i,k} \neq \text{sign}(g_k) \mid n_i) \\
&\overset{(a)}{\leq} \sum_{n_i:|n_i-np|\geq n\varepsilon} \mathbb{P}(n_i) + \mathbb{P}(c_{i,k} \neq \text{sign}(g_k) \mid n_i = n(p-\varepsilon)) \\
&\overset{(b)}{\leq} 2\mathrm{e}^{-2\varepsilon^2 n} + \mathrm{e}^{-n(p-\varepsilon)\frac{S_k^2}{2(S_k^2+4)}} \leq \frac{2}{n^{2C}} + \mathrm{e}^{-\sqrt{Cn\log(n)}\frac{S_k^2}{2(S_k^2+4)}} \leq q_k^\star,
\end{aligned}
$$

where (a) is from the fact that the upper bound in (E.1) is a decreasing function of $n_i$, and (b) holds from Lemma E.1 and Lemma C.5, the Hoeffding's inequality on the binomial distribution.

### E.2   Proof of Lemma 2

Let an attack vector $\boldsymbol{\beta}$ and an attack function $f_{\boldsymbol{\beta}}(\cdot)$ given. Consider an arbitrary $\boldsymbol{m} \in M^+$. From the definitions of $\mu$ and $\hat{\mu}$, we have $\mu = \hat{\mu}$ iff $f_{\boldsymbol{\beta}}(\phi(\boldsymbol{m})) \in Y^+$. Similarly, for an arbitrary $\boldsymbol{m} \in M^-$, we have $\mu = \hat{\mu}$ iff $f_{\boldsymbol{\beta}}(\phi(\boldsymbol{m})) \in Y^-$. Thus, from the definitions of $Y^+$ and $Y^-$, the sufficient and necessary condition for $b-$Byzantine tolerance can be expressed as follows.

**Proposition 2.** *The perfect $b-$Byzantine tolerance condition is equivalent to the following:* $\forall \boldsymbol{\beta} \in B_b, \forall f_{\boldsymbol{\beta}} \in \mathcal{F}_{\boldsymbol{\beta}}$,

$$
\begin{cases}
\|f_{\boldsymbol{\beta}}(\phi(\boldsymbol{m}))\|_0 > \left\lfloor \frac{n}{2} \right\rfloor, & \forall \boldsymbol{m} \in M^+ \\
\|f_{\boldsymbol{\beta}}(\phi(\boldsymbol{m}))\|_0 \leq \left\lfloor \frac{n}{2} \right\rfloor, & \forall \boldsymbol{m} \in M^-
\end{cases}
\tag{E.2}
$$

The condition stated in Proposition 2 can be further simplified as follows.

**Proposition 3.** *The perfect $b-$Byzantine tolerance condition in Proposition 2 is equivalent to*

$$
\begin{cases}
\|\phi(\boldsymbol{m})\|_0 > \left\lfloor \frac{n}{2} \right\rfloor + b, & \forall \boldsymbol{m} \in M^+ \\
\|\phi(\boldsymbol{m})\|_0 \leq \left\lfloor \frac{n}{2} \right\rfloor - b, & \forall \boldsymbol{m} \in M^-
\end{cases}
\tag{E.3}
$$

*Proof.* Consider arbitrary $\boldsymbol{m} \in M^-$. We want to prove that

$$
\forall \boldsymbol{\beta} \in B_b, \forall f_{\boldsymbol{\beta}} \in \mathcal{F}_{\boldsymbol{\beta}}, \quad \|f_{\boldsymbol{\beta}}(\phi(\boldsymbol{m}))\|_0 \leq \left\lfloor \frac{n}{2} \right\rfloor
\tag{E.4}
$$

is equivalent to

$$
\|\phi(\boldsymbol{m})\|_0 \leq \left\lfloor \frac{n}{2} \right\rfloor - b.
\tag{E.5}
$$

First, we show that (E.5) implies (E.4). According to Lemma C.1 $\|f_{\boldsymbol{\beta}}(\phi(\boldsymbol{m})) \oplus \phi(\boldsymbol{m})\|_0 \leq b$ holds for arbitrary $\boldsymbol{\beta} \in B_b$ and arbitrary $f_{\boldsymbol{\beta}} \in \mathcal{F}_{\boldsymbol{\beta}}$. Thus,

$$
\|f_{\boldsymbol{\beta}}(\phi(\boldsymbol{m}))\|_0 \leq \|f_{\boldsymbol{\beta}}(\phi(\boldsymbol{m})) \oplus \phi(\boldsymbol{m})\|_0 + \|\phi(\boldsymbol{m})\|_0 \leq b + \left( \left\lfloor \frac{n}{2} \right\rfloor - b \right) = \left\lfloor \frac{n}{2} \right\rfloor
$$

holds for $\forall \boldsymbol{\beta} \in B_b, \forall f_{\boldsymbol{\beta}} \in \mathcal{F}_{\boldsymbol{\beta}}$, which completes the proof. Now, we prove that (E.4) implies (E.5), by contra-position. Suppose $\|\phi(\boldsymbol{m})\|_0 > \left\lfloor \frac{n}{2} \right\rfloor - b$. We divide the proof into two cases. The first case is when $\|\phi(\boldsymbol{m})\|_0 > n - b$. In this case, we arbitrary choose $\boldsymbol{\beta}^* \in B_b$ and select the identity mapping $f_{\boldsymbol{\beta}^*}^* : \boldsymbol{c} \mapsto \boldsymbol{y}$ such that $y_j = c_j$ for all $j \in [n]$. Then, $\|f_{\boldsymbol{\beta}^*}^*(\phi(\boldsymbol{m}))\|_0 = \|\phi(\boldsymbol{m})\|_0 > n - b \geq n - \lfloor n/2 \rfloor \geq \lfloor n/2 \rfloor$. Thus, we can state that

$$
\exists \boldsymbol{\beta}^* \in B_b, \exists f_{\boldsymbol{\beta}^*}^* \in \mathcal{F}_{\boldsymbol{\beta}^*} \text{ such that } \|f_{\boldsymbol{\beta}^*}^*(\phi(\boldsymbol{m}))\|_0 \leq \left\lfloor \frac{n}{2} \right\rfloor
$$

when $\|\phi(\boldsymbol{m})\|_0 > n - b$, which completes the proof for the first case. Now consider the second case where $\lfloor n/2 \rfloor - b < \|\phi(\boldsymbol{m})\|_0 \leq n - b$. To begin, denote $\phi(\boldsymbol{m}) = \boldsymbol{c} = [c_1, c_2, \cdots, c_n]$. Let

$S = \{i \in [n] : c_i = 0\}$, and select $\boldsymbol{\beta}^* \in B_b$ which satisfies[6] $\{i \in [n] : \beta_i^* = 1\} \subseteq S$. Now define $f_{\boldsymbol{\beta}^*}^*(\cdot)$ as $f_{\boldsymbol{\beta}^*}^*(\phi(\boldsymbol{m})) = \phi(\boldsymbol{m}) \oplus \boldsymbol{\beta}^*$. Then, we have

$$\|f_{\boldsymbol{\beta}^*}^*(\phi(\boldsymbol{m}))\|_0 = \|\phi(\boldsymbol{m})\|_0 + \|\boldsymbol{\beta}^*\|_0 > \left\lfloor \frac{n}{2} \right\rfloor - b + b = \left\lfloor \frac{n}{2} \right\rfloor.$$

Thus, the proof for the second case is completed, and this completes the statement of (E.3) for arbitrary $\boldsymbol{m} \in M^-$. Similarly, we can show that

$$\forall \boldsymbol{\beta} \in B_b, \forall f_{\boldsymbol{\beta}} \in \mathcal{F}_{\boldsymbol{\beta}}, \quad \|f_{\boldsymbol{\beta}}(\phi(\boldsymbol{m}))\|_0 > \left\lfloor \frac{n}{2} \right\rfloor$$

is equivalent to $\|\phi(\boldsymbol{m})\|_0 > \left\lfloor \frac{n}{2} \right\rfloor + b$ for arbitrary $\boldsymbol{m} \in M^+$. This completes the proof. □

Now, we further reduce the condition in Proposition 3 as follows.

**Proposition 4.** *The perfect $b-$Byzantine tolerance condition in Proposition 3 is equivalent to*

$$\|\phi(\boldsymbol{m})\|_0 \leq \left\lfloor \frac{n}{2} \right\rfloor - b, \quad \forall \boldsymbol{m} \in M^- \tag{E.6}$$

*Proof.* All we need to prove is that (E.6) implies (E.3). Assume that the mapping $\phi$ satisfies (E.6). Consider an arbitrary $\boldsymbol{m}' \in M^+$ and denote $\boldsymbol{m}' = [m_1', m_2', \cdots, m_n']$. Define $\boldsymbol{m} = [m_1, m_2, \cdots, m_n]$ such that $m_i' \oplus m_i = 1$ for all $i \in [n]$. Then, we have $\boldsymbol{m} \in M^-$ from the definitions of $M^+$ and $M^-$. Now we denote $\phi(\boldsymbol{m}) = \boldsymbol{c} = [c_1, c_2, \cdots, c_n]$ and $\phi(\boldsymbol{m}') = \boldsymbol{c}' = [c_1', c_2', \cdots, c_n']$. Then, $c_j \oplus c_j' = 1$ holds for all $j \in [n]$ since $E_j(\cdot)$ is a majority vote function[7]. In other words, $\|\phi(\boldsymbol{m})\|_0 + \|\phi(\boldsymbol{m}')\|_0 = n$ holds. Thus, if a given mapping $\phi$ satisfies $\|\phi(\boldsymbol{m})\|_0 \leq \lfloor n/2 \rfloor - b$ for all $\boldsymbol{m} \in M^-$, then $\|\phi(\boldsymbol{m}')\|_0 \geq n - (\lfloor n/2 \rfloor - b) = \lceil n/2 \rceil + b > \lfloor n/2 \rfloor + b$ holds for all $\boldsymbol{m} \in M^+$, which completes the proof. □

In order to prove Lemma 2, all that remains is to prove that (E.6) reduces to

$$\sum_{v=1}^{\lfloor n/2 \rfloor} |S_v(\boldsymbol{m})| \leq \lfloor n/2 \rfloor - b \qquad \forall \boldsymbol{m} \in M_{\lfloor n/2 \rfloor}. \tag{E.7}$$

Recall that $\phi(\boldsymbol{m}) = \boldsymbol{c} = [c_1, c_2, \cdots, c_n]$ where $c_j = \text{maj}(\{m_i\}_{i \in P_j})$ and $P_j = \{i \in [n] : G_{ji} = 1\}$. Moreover, we assumed that $|P_j| = \|\mathbf{G}(j,:)\|_0$ is an odd number. Thus, $c_j = \mathbb{1}_{\{\|\mathbf{G}(j,:)\|_0 + 1 \leq 2\boldsymbol{m}^T \mathbf{G}(j,:)\}}$, and the set $[n] = \{1, 2, \cdots, n\}$ can be partitioned as $[n] = S_1 \cup S_2 \cup \cdots \cup S_{\lfloor n/2 \rfloor + 1}$ where $S_v := \{j \in [n] : \|\mathbf{G}(j,:)\|_0 = 2v - 1\}$. Therefore, for a given $\boldsymbol{m} \in M^-$, we have

$$\|\phi(\boldsymbol{m})\|_0 = \sum_{j=1}^{n} c_j = \sum_{v=1}^{\lfloor n/2 \rfloor + 1} |\{j \in S_v : c_j = 1\}|$$

$$= \sum_{v=1}^{\lfloor n/2 \rfloor + 1} \left| \left\{ j \in S_v : \boldsymbol{m}^T \mathbf{G}(j,:) \geq \frac{\|\mathbf{G}(j,:)\|_0 + 1}{2} + 1 = v \right\} \right| = \sum_{v=1}^{\lfloor n/2 \rfloor + 1} |S_v(\boldsymbol{m})|.$$

Note that $S_v(\boldsymbol{m})$ for $v = \lfloor n/2 \rfloor + 1$ reduces to

$$S_{\lfloor n/2 \rfloor + 1}(\boldsymbol{m}) = \{j \in [n] : \|\mathbf{G}(j,:)\|_0 = 2(\lfloor n/2 \rfloor - 1) + 1, \boldsymbol{m}^T \mathbf{G}(j,:) \geq \lfloor n/2 \rfloor + 1\} = \varnothing$$

since $\boldsymbol{m} \in M^-$. Thus, combining the two equations above, we obtain the following.

**Proposition 5.** *The perfect $b-$Byzantine tolerance condition in Proposition 4 is equivalent to*

$$\sum_{v=1}^{\lfloor n/2 \rfloor} |S_v(\boldsymbol{m})| \leq \left\lfloor \frac{n}{2} \right\rfloor - b \qquad \forall \boldsymbol{m} \in M^-,$$

*or equivalently,*

$$\sum_{v=1}^{\lfloor n/2 \rfloor} |S_v(\boldsymbol{m})| \leq \left\lfloor \frac{n}{2} \right\rfloor - b \qquad \forall \boldsymbol{m} \in M_t, \quad \forall t = 0, 1, \cdots, \lfloor n/2 \rfloor. \tag{E.8}$$

Now, we show that (E.8) is equivalent to (E.7). We can easily check that the former implies the latter, which is directly proven from the statements. Thus, all we need to prove is that (E.7) implies (E.8). First, when $t = 0$, note that $|S_v(\boldsymbol{m})| = 0$ for $\forall \boldsymbol{m} \in M_0, \forall v \in \{1, 2, \cdots, \lfloor n/2 \rfloor\}$, which implies that (E.8) holds trivially. Thus, in the rest of the proof, we assume that $t > 0$.

Consider an arbitrary $t \in \{1, 2, \cdots \lfloor n/2 \rfloor\}$ and an arbitrary $\boldsymbol{m} \in M_t$. Denote $\boldsymbol{m} = \boldsymbol{e}_{i_1} + \boldsymbol{e}_{i_2} + \cdots + \boldsymbol{e}_{i_t}$ where $\boldsymbol{e}_1 = [1, 0, \cdots, 0]$, $\boldsymbol{e}_2 = [0, 1, 0, \cdots, 0]$, and $\boldsymbol{e}_n = [0, \cdots, 0, 1]$. Moreover, consider an arbitrary $\boldsymbol{m}' \in M_{\lfloor n/2 \rfloor}$ which satisfies $m'_i = 1$ for $i = i_1, i_2, \cdots, i_t$. Denote $\boldsymbol{m}' = \boldsymbol{e}_{i_1} + \cdots + \boldsymbol{e}_{i_t} + \boldsymbol{e}_{j_1} + \cdots + \boldsymbol{e}_{j_{\lfloor n/2 \rfloor - t}}$. Then, $(\boldsymbol{m}' - \boldsymbol{m})^T \mathbf{G}(j, :) \geq 0$ holds for all $j \in [n]$, which implies $S_v(\boldsymbol{m}) \subseteq S_v(\boldsymbol{m}')$ for all $v = 1, 2, \cdots, \lfloor n/2 \rfloor$. Thus, we have $|S_v(\boldsymbol{m})| \leq |S_v(\boldsymbol{m}')|$ for all $v \in \{1, 2, \cdots, \lfloor n/2 \rfloor\}$, which implies $\sum_{v=1}^{\lfloor n/2 \rfloor} |S_v(\boldsymbol{m})| \leq \sum_{v=1}^{\lfloor n/2 \rfloor} |S_v(\boldsymbol{m}')|$. Since this holds for arbitrary $\boldsymbol{m}' \in M_{\lfloor n/2 \rfloor}$, $\boldsymbol{m} \in M_t$, and $t \in \{1, 2, \cdots, \lfloor n/2 \rfloor\}$, we can conclude that (E.7) implies (E.8). All in all, (E.8) is equivalent to (E.7). Combining this with Propositions 2,3, 4 and 5 completes the proof of Lemma 2.

### E.3 Proof of Lemma E.1

From Lemmas 1 and C.4, we have

$$q_k^{(idv)} = \mathbb{P}(\text{sign}(\tilde{g}_k^{(j)}) \neq \text{sign}(g_k)) \leq \begin{cases} \frac{2}{9} \frac{1}{S_k^2} & \text{if } S_k > \frac{2}{\sqrt{3}} \\ \frac{1}{2} - \frac{S_k}{2\sqrt{3}} & \text{otherwise} \end{cases} \tag{E.9}$$

for arbitrary $j \in [n], k \in [d]$ where $S_k = \frac{|g_k|}{\bar{\sigma}_k}$ is defined in Definition 1. Denote the set of data partitions assigned to node $i$ by $P_i = \{j_1, j_2, \cdots, j_{n_k}\}$. Define a random variable $X_s$ as

$$X_s = \mathbb{1}_{\text{sign}(\tilde{g}_k^{(j_s)}) = \text{sign}(g_k)}.$$

Then, from the definition of $q_k^{(idv)}$ in (E.9), we have $\mathbb{P}(X_s = 1) = p_k^{(idv)} := 1 - q_k^{(idv)}$, and $\mathbb{P}(X_s = 0) = q_k^{(idv)}$. Recall that $c_{i,k} = \text{maj}\{\text{sign}(\tilde{g}_k^{(j)})\}_{j \in P_i}$. By using a new random variable defined as $X := \sum_{s=1}^{n_i} X_s$, the failure probability of node $i$ estimating the sign of $g_k$ is represented as

$$\mathbb{P}(c_{i,k} \neq \text{sign}(g_k)|n_i) = \mathbb{P}(X \leq \frac{n_i}{2}) = \mathbb{P}(X - n_i p_k^{(idv)} \leq -n_i(-\frac{1}{2} + p_k^{(idv)}))$$

$$\overset{(a)}{\leq} e^{-2(-\frac{1}{2} + p_k^{(idv)})^2 n_i} \overset{(b)}{\leq} e^{-n_i \frac{S_k^2}{2(S_k^2 + 4)}}$$

where (a) is from Lemma C.5 and (b) is from the fact that $\frac{1}{4\left(-\frac{1}{2} + p_k^{(idv)}\right)^2} - 1 \leq \frac{4}{S_k^2}$, which is shown as below. Note that

$$-\frac{1}{2} + p_k^{(idv)} = \frac{1}{2} - q_k^{(idv)} \geq \begin{cases} \frac{1}{2} - \frac{2}{9} \frac{1}{S_k^2} & \text{if } S_k > \frac{2}{\sqrt{3}} \\ \frac{S_k}{2\sqrt{3}} & \text{otherwise} \end{cases}$$

When $S_k \leq \frac{2}{\sqrt{3}}$, we have $\frac{1}{4\left(-\frac{1}{2} + p_k^{(idv)}\right)^2} - 1 \leq \frac{3}{S_k^2} - 1 < \frac{4}{S_k^2}$. For the case of $S_k > \frac{2}{\sqrt{3}}$, we have

$$\frac{1}{4\left(-\frac{1}{2} + p_k^{(idv)}\right)^2} - 1 \leq \frac{1}{S_k^2} \frac{\frac{8}{9} - \frac{16}{81} \frac{1}{S_k^2}}{1 - \frac{8}{9} \frac{1}{S_k^2} + \frac{16}{81} \frac{1}{S_k^4}} < \frac{1}{S_k^2} \frac{\frac{8}{9}}{1 - \frac{8}{9} \frac{1}{S_k^2}} < \frac{4}{S_k^2}$$

where the last inequality is from the condition on $S_k$. This completes the proof of Lemma E.1.

### E.4 Proof of Proposition 1

Let $u_k^{(j)} = \tilde{g}_k^{(j)} - g_k$. From the definition of $\tilde{g}_k^{(j)}$, we have $Y = Bu_k^{(j)} = B(\tilde{g}_k^{(j)} - g_k) = \sum_{x \in B_j} u_k(x)$. From Lemma C.2, $f_Y = \text{conv}\{f_{u_k(x)}\}_{x \in B_j}$. Since $u_k(x)$ are zero-mean, symmetric, and unimodal from Assumption 4, Lemma C.3 implies that $Y$ (and thus $u_k^{(j)}$) is also zero-mean, symmetric, and unimodal. Therefore, $\tilde{g}_k^{(j)} = g_k + u_k^{(j)}$ is unimodal and symmetric around the mean $g_k$. The result on the variance of $\tilde{g}_k^{(j)}$ is directly obtained from the independence of $\tilde{g}_k(x)$ for different $x \in B_j$.

## Footnotes

[4]Note that (D.14) implies (D.13), when the condition part is restricted to $|S_{b+1}(\boldsymbol{m})| = q$.

[5]In Fig. D.7, one can confirm that $S_{b+1}(\boldsymbol{m})$ is not consisted of consecutive integers (i.e., there's a gap), while $S_{b+1}(\boldsymbol{m}'')$ has no gap. Thus, we call this process as *gathering* process.

[6] We can always find such $\boldsymbol{\beta}^*$ since $|S| \geq b$ due to the setting of $\|\phi(\boldsymbol{m})\|_0 \leq n - b$.

[7] Recall that $c_j = E_j(\{m_i\}_{i \in P_j}) = \text{maj}(\{m_i\}_{i \in P_j})$ and $c_j' = \text{maj}(\{m_i'\}_{i \in P_j})$. Thus, $m_i' \oplus m_i = 1$ for all $i \in [n]$ implies that $c_j \oplus c_j' = 1$ holds for all $j \in [n]$.