[Reviews · NeurIPS 2020]

Review 1

Summary and Contributions: This paper addresses the problem of designing first-order optimization methods that are both communication efficient and robust to byzantine workers. In particular, the paper focuses on an existing variant of SignSGD, namely SignSGD with majority voting (SignSGD-MV), which is already communication efficient by design. The paper proposes a new coding theoretic approach to make SignSGD-MV robust to byzantine workers. In a regular SignSGD-MV method, each of the n workers computes a gradient estimate based on the data partition assigned to it and sends a sign of the gradient estimate to the master node. The master node takes the coordinate wise majority of the signed gradient estimates received from all the workers to obtain the final signed gradient estimate. However, the byzantine workers may send adversarially generated signed gradient estimates so that the majority operation at the master end up computing the incorrect final signed gradient estimate. This paper proposed a novel election coding framework where each worker node is assigned multiple data partitions. Each worker then sends the coordinate wise majority of the local signed gradient estimates based on its partitions to the master. The master then performs the coordinate wise majority of the signed vectors (outputs of the local majority operations) received from the workers. The paper shows that with a careful design of the data partition allocation to the workers can ensure that the master computes the correct final signed gradient estimate despite the presence of the byzantine workers. The paper studies two allocation schemes (coding schemes): 1) random Bernoulli codes that provide byzantine tolerance with high probability, and 2) deterministic codes that provide perfect byzantine tolerance. The paper then evaluates the proposed coding schemes on Amazon EC2. For two combinations of datasets and model architecture, the paper empirically shows that the proposed schemes significantly improve upon the regular (uncoded) SignSGD-MV. ######### Post author response phase ########### Thank you for taking the time to address my comments. I have updated my score accordingly. I would encourage the authors to incorporate their responses and new results (from their response) to the main body of the revised version.

Strengths: The paper studies a timely problem of designing communication efficient and robust distributed optimization methods. The paper proposes the novel framework of election coding and presents two coding schemes under this framework along with their theoretical analysis. The paper carries out experiments on the real systems and demonstrates the utility of the proposed coding schemes over the natural baseline of regular (uncoded) SignSGD-MV.

Weaknesses: The proposed scheme significantly increases the amount of redundant computation performed at the workers. The authors should discuss this point in detail in the main text. For example, for the deterministic scheme, if one takes b=1, Theorem 3 shows that, for large n, computational redundancy becomes, r = \Omega(n/4), which is rather large. Would it be more beneficial to use full other robust schemes based on full gradients + median approaches? The paper also does not address the issue of how far the proposed schemes are from optimal solutions. The experiment results can be more comprehensive. Please include the results for *no_attack* in Fig. 5 and 6. Also, the author should present results for larger networks. For such networks, with large n, do the proposed scheme have tolerable computational redundancy? The paper only considers the setting where n data partitions are assigned to n workers. Could the authors comment on the case where # of data partitions is different from # of workers? Is it possible to lower the computational redundancy by exploiting this dimension?

Correctness: The results in the paper appear correct. (The reviewer did not verify the details of the proof.)

Clarity: The paper is generally well written. The reviewer has some minor comments/suggestions to improve the quality/clarity of the presentation. Please see the weaknesses and the additional feedback sections.

Relation to Prior Work: The paper adequately covers the relevant prior work. In information theory and complexity theory literature, there has been some work on understanding the boolean computation in the presence of noisy gates. Could the authors comment on the relevance of that literature to their work?

Reproducibility: Yes

Additional Feedback: 1. For deterministic scheme, in Lemma 2, the authors only focus on the vectors m with ||m||_0 = \lceil n/2 \rceil. Clearly by the definition of S_v(m) in (4), master would obtain the correct result for ||m||_0 < \lceil n/2 \rceil. Does this also hold for the ||m||_0 < \lceil n/2 \rceil case? 2. In equation (2), q --> q_i and S --> S_i ? 3. The coding scheme in Figure 1(b) and 2 is not tolerant to 1 byzantine worker. As shown in Fig. 1(b) (D1, D2, D3, D4, D5) = (-1, -1, +1, -1, +1) gives (c1, c2, c3, c4, c5) = (-1, -1, +1, -1, -1), which ensures that the master receives three -1 even in the presence of 1 byzantine. However, (D1, D2, D3, D4, D5) = (-1, +1, -1, -1, +1) gives (c1, c2, c3, c4, c5) = (-1, +1, -1, +1, -1). Here, clearly 1 byzantine would lead to an error at the master. Please consider presenting an example that is actually tolerant to 1 Byzantine worker. 4. In line 83-85, "The codes proposed in [10] guarantees Byzantine tolerance, as long as each node transmits real-valued gradients, while our codes provide tolerance even under the compressed gradient constraint." This gives the impression that the proposed codes work for both settings, with the compressed gradients, and with the real gradients. I would be happy to revise my score based on the author's responses to my comments.


Review 2

Summary and Contributions: In this work the authors propose ELECTION CODING which is byzantine robust and communication efficient in distributed learning framework. In this framework with n worker nodes, the data is split into n blocks. The authors propose randomized (Bernoulli coding ) and a deterministic data allocation scheme over the workers. The sign of the vector is used to provide low communication overhead and majority voting scheme along with data redundancy using randomized and deterministic allocation matrix is for robustness of the algorithm. The paper also offers analysis and experimental result in the support of the algorithm.

Strengths: 1. In terms of redundancy this paper improves over the previous paper DRACO. 2.The hierarchical voting scheme seems like a good idea to handle the error in mismatch of the sign of the gradient locally and globally. 3. The analysis for random Bernoulli code and design of deterministic codes (as claimed by the authors) looks promising. 4. The experimental result shows improvement over the previous work of signsgd with majority voting.

Weaknesses: Please see the comments.

Correctness: I have not checked the proof for the section 4 (deterministic code). The rest of the paper looks ok.

Clarity: The paper is overall well-written.

Relation to Prior Work: The related work section is very concise and addresses the relevant works in quantization and byzantine robust learning.

Reproducibility: No

Additional Feedback: 1. The byzantine setup seems a bit weak as the byzantine attacks only happen by bit flipping. The paper does not address the problem when the attack happens in computing gradient. For example if Gaussian noise is added to the gradient of a data block and then the sign of the gradient is applied. 2. With the data allocation scheme, the computational load increases. Also the computational load is not uniform among the workers. 3. The idea of the paper seems to be the fact that in absence of the byzantine set-up it should be sign-sgd algorithm. The sign-sgd algorithm uses the majority voting of the signs of the gradients form workers. The lemma 1 focuses on the local error which is probability of error in the mismatch of the sign of g and c_i. Should we not worry about the sign of the g_i (local gradient)? ######################### The authors have responded well to all my queries in the review and accordingly I am changing my score to 6.


Review 3

Summary and Contributions: This paper proposes a Byzantine-robust framework for signed stochastic gradient descent (SignSGD) based on the SignSGD majority vote framework. Two election coding schemes are constructed and experiments are done to confirm the mathematical results. ----- Post rebuttal: After reading the authors' response, the authors do address my concern about where the errors could be introduced (although I still think the way it is worded in the paper could lead to misunderstanding this point). However, I still question the novelty of the methods. I will keep my score.

Strengths: The paper supports the claims made with proofs and experimental results. The problem studied is significant and valuable.

Weaknesses: The paper proposes a scheme, based on the majority voting scheme for the SignSGD, however, the solution is not so novel. Moreover, the discussion of the scheme is a little confusing and a simple example of how the algorithm works exactly would benefit the paper largely. Also, in Figure 1b, when the bit flip happened between the polling station and the master, that did not affect the results, but if the bit flip has happened between the voters and the polling stations, then it would have changed enough of the polling station results to change the final result. In remark 1, you say that the transmitted messages of the nodes are compromised, so that would mean that the change could happen between the node and the polling stations. The byzantine attack happening only between the polling stations and the master needs more justification.

Correctness: The paper satisfies the claims and the methodology is correct.

Clarity: The paper is well written, however, it could benefit from an explicit example and some more high level explanation of the methods.

Relation to Prior Work: Prior work is discussed clearly, and the links between this work and the preceding literature is well explained.

Reproducibility: Yes

Additional Feedback:

[Author Response · NeurIPS 2020]

We thank the reviewers for all remarks/suggestions. We appreciate the acknowledgments on: problem formulation
being timely (**R1**) and significant/valuable (**R3**); extensive analysis and experimental results provided (**R1**, **R2**, **R3**);
novelty (**R1**) and improved redundancy over existing DRACO (**R3**); and good utilization of hierarchical voting (**R2**).
As for the concerns/questions raised, we believe we successfully addressed every single one, as explained below.
**Computational load (R1, R2)** True, any coding requires redundant computation in return for better Byzantine
protection. Our method offers great options to protect system with reasonable extra computation. As an example, our
Bernoulli code with mean redundancy of 2 (two partitions per worker), a reasonable redundancy factor, achieves an
88% accuracy at unit time of 609 while uncoded SignSGD-MV reaches a 79% accuracy at a slower time of 750 and
fails altogether to reach the 88% level with 3 out of 9 workers compromised (Table 1 in Supplementary Materials (SM)).
Further, using reduced minibatch size, effective redundancy reduces to as low as 1.25, for example, while providing a
solid 85% accuracy for a severe 40% Byzantine scenario (2 out of 5 compromised) as in Fig. A.3a of SM. Under the
same condition, uncoded SignSGD-MV fails to give over a 40% accuracy, regardless of train time. Reducing minibatch
size for uncoded scheme results in unacceptable performance (note that for our method, each worker extracts gradient
from multiple data partitions so that the detrimental effect of reducing minibatch size is not felt in a significant way).
**Additional experiments (R1, R2)** We obtained new experimental results on full gradient + median (FGM) (Fig. 1a
below) and also for no-attack scenarios for the severe 40% Byzantine case (Fig. 1abc). Our scheme with redundancy
as low as $r_{\text{eff}} = 1.5$ outperforms FGM. While FGM requires no computational redundancy, it needs 32x more
communication bandwidth than ours while performing worse. For the 20% Byzantine scenario ($b = 3$), FGM performs
relatively better, but still falls well below ours. For the severe 40% Byzantine case, it is harder to achieve near-perfect
protection but our schemes clearly perform better. We also ran experiments for a larger network, ResNet-50, as seen
in Fig. 1b. Our scheme with redundancy $r_{\text{eff}} = 1.5$ is still effective here. Thanks to the suggestion by **R2**, we added
independent Gaussian noise to all gradients corresponding to individual partitions before the signs are taken (in addition
to Byzantine attacks on local majority signs). The proposed Bernoulli code can tolerate noisy gradient computation
well, while uncoded signSGD-MV cannot, as shown in Fig. 1c for added noise with variance $\sigma^2 = 1e^{-8}$.
**Optimal? (R1)** This is a tough but highly relevant question. Rephrased, we ask: what is minimal code redundancy
while providing (asymptotically) perfect Byzantine tolerance? From L273 of SM, minimum connection probability
(proportional to redundancy) of random codes for full Byzantine protection must satisfy $p = f(k)/\sqrt{k}$ with $f(k)$ being
a growing function of $k$, # of data partitions. Our choice $p \sim \sqrt{\log(k)/k}$ meets this, and thus is asymptotically optimal.
**What if # of data partitions $k$ is not equal to # of workers $n$ (R1)** An interesting question. When $n \neq k$, the required
connection probability for Byzantine tolerance in Lemma 1 behaves as $p \sim \sqrt{\log(k)/k}$; computational load at each
worker is proportional to $p$ and thus load can be lowered by increasing $k$. However, the local majority vote becomes
less accurate with increasing $k$ since each worker uses fewer data points (parameter $S$ deteriorates; see $S$ in equation 2
and Proposition/Definition 1). We feel a more effective solution is reducing mini-batch size as shown in A.4 in SM.
**Versus Boolean computation (R1)** Boolean computation on noisy gates optimizes # of gates for accuracy; we concern
how to connect inputs to majority gates to tolerate Byzantines with minimal connections, an entirely different issue.
**Clarifications (R1, R2)** Yes, the master obtains the correct result for $\|\boldsymbol{m}\|_0 < \lfloor n/2 \rfloor$, according to proof of Lemma 2
(**R1**). Our intention in Lemma1 is to ensure $c_i$, the local majority vote, closely reflects sign of global gradient $g$ (**R2**).
**Minor correction (R1)** Our scheme focuses only on quantized gradients; we will correct this and typos in equation (2).
**Nonuniform computational load (R2)** True, our worker loads are non-uniform, but load distribution for Bernoulli
codes tends to uniform as $n$ grows, due to concentration property of binomial distribution. The probability that
individual load deviates from mean $np$ by $n\varepsilon$ is less than $2e^{-2\varepsilon^2 n}$, as seen in Lemma C.5 of SM.
**Solution not so novel (R3)** We disagree. If **R3** implies majority voting is the gist of our method, we stress it is not. In
any case, Election Coding is about redundant allocation of data partitions to workers in conjunction with hierarchical
voting, a new framework that does not exist in coding/information theory community or machine learning community.
**Attacks between voters and polling station (R3)** No attacks can take place between voters and polling stations. Think
of the voters (and their clones) actually going to the polling stations and casting ballots themselves with no possibility
for voting fraud up to that point. In the real setup, voters are like data partitions while polling stations represent workers.
The individual gradients are actually computed at the worker nodes with no exposed connections. See Fig. 1d below.
**High level explanation and better example (R3, R1)** Figs. 1&2 along with associated discussions provide high level
understanding on suggested coding methodology for redundant distribution of data partitions to workers (also see
Fig. 1d below for a new example); perhaps what **R3** is asking is a high-level description of distributed learning itself.
It will be a simple thing to add this in the revision, given a chance. Thanks to **R1**'s request, Fig. 1d below gives an
example code that perfectly tolerates $b = 1$ Byzantine on any node. We will switch to this example in the revision.

(a) Compare with median     (b) ResNet-50     (c) Noisy computation     (d) Perfect Byzantine-tolerant example

Figure 1: (a)-(c): Experimental results on CIFAR-10 dataset, for $n = 15$ and $b = 6$ (unless stated otherwise), averaged out over three trials; (d): Example of the suggested deterministic code that perfectly tolerates $b = 1$ Byzantine out of $n = 5$ nodes

[Meta-Review · NeurIPS 2020]

Overall, reviewers found this a solid contribution to the goal of designing communication efficient and robust distributed optimization methods. They continued to have questions about the significance and scope of the solution, but appreciated the authors' analysis and clarity, both in the paper and in the author response. We also thank the authors for the clear description of the NeurIPS 2019 reviews, which helped us appreciate the many problems solved in this research and that another set of reviews had already largely been addressed. Please read the reviews carefully and incorporate these comments into the final version as best you can.